# FAKE IT TILL YOU MAKE IT: TOWARDS ACCURATE NEAR-DISTRIBUTION NOVELTY DETECTION

**Hossein Mirzae**[1], **Mohammadreza Salehi**[2], **Sajjad Shahabi** [1], **Efstratios Gavves**[2],
**Cees G. M. Snoek**[2], **Mohammad Sabokrou**[3][4], **Mohammad Hossein Rohban**[1]
[1] Sharif University of Technology [2] University of Amsterdam
[3] Institute for Research in Fundamental Sciences (IPM)
[4] Okinawa Institute of Science and Technology

## ABSTRACT

We aim for image-based novelty detection. Despite considerable progress, existing models either fail or face a dramatic drop under the so-called "near-distribution" setting, where the differences between normal and anomalous samples are subtle. We first demonstrate existing methods experience up to 20% decrease in performance in the near-distribution setting. Next, we propose to exploit a score-based generative model to produce synthetic near-distribution anomalous data. Our model is then fine-tuned to distinguish such data from the normal samples. We provide a quantitative as well as qualitative evaluation of this strategy, and compare the results with a variety of GAN-based models. Effectiveness of our method for both the near-distribution and standard novelty detection is assessed through extensive experiments on datasets in diverse applications such as medical images, object classification, and quality control. This reveals that our method considerably improves over existing models, and consistently decreases the gap between the near-distribution and standard novelty detection performance. The code repository is available at https://github.com/rohban-lab/FITYMI.

## 1 INTRODUCTION

In novelty detection (ND)[1], the goal is to learn to identify test-time samples that unlikely to come from the training distribution, without having access to any class label data of the training set (40). Such samples are called anomalous, while the training set is referred to as normal. One has access to only normal data during training in ND. Recently, PANDA (34) and CSI (44) have considerably pushed state-of-the-art and achieved more than 90% the area under the receiver operating characteristics (AUROC) on the CIFAR-10 dataset (23) in the ND task, where one class is assumed to be normal and the rest are considered anomalous. However, as we will show empirically, these methods struggle to achieve a similar performance in situations where outliers are semantically close to the normal distribution, e.g. instead of distinguishing dog vs. car, which is the regular ND, one desires to distinguish dog vs. fox in such settings. That is, they experience a performance drop when faced with such near-anomalous inputs. In this paper, our focus is on such scenarios, which we call near novelty detection (near-ND). We note that near-ND is a more challenging task and has been explored to a smaller extent. Near novelty detection has found several important practical applications in diverse areas such as medical imaging, and face liveness detection (31).

Our first contribution is to benchmark eight recent novelty detection methods in the near-ND setting, which consists of ND problems whose normal and anomaly classes are either naturally semantically close, or else synthetically forced to be close. Fig. 1 compares the performance of PANDA (34) and CSI (44) in an instance of the near-ND, and the standard ND setups, which shows roughly a 20% AUROC drop in near-ND compared with the ND. Furthermore, while MHRot (16) performs relatively comparable to PANDA and CSI in ND, it is considerably worse in near-ND, highlighting the need for near novelty detection benchmarking.

---

[1] In the literature novelty detection and anomaly detection are used interchangeably. We use the term novelty detection (ND) throughout this paper.

A similar problem setup has recently been investigated in the out-of-distribution (OOD) detection domain, known as "near out-of-distribution" detection (9), where the in-distribution and out-of-distribution samples are semantically similar. OOD detection and ND are closely related problems with the primary difference being that unlike OOD detection, the labels for sub-classes of the normal data are not accessible during training in ND, i.e. if the normal class is car, the type of car is given for each normal sample during training in OOD detection, while being unknown in ND. This makes anomaly detection a more challenging problem than the OOD detection, as this side information turns out to be extremely helpful in uncertainty quantification (45). To cope with the challenges in the near-OOD detection, (9), (16), and (36) employ outlier exposure techniques, i.e., exposing the model to the real outliers, available on the internet, during training. Alternatively, some approaches (21; 32) utilized GANs to generate outliers. Such real or synthetic outliers are used in addition to the normal data to train the OOD detection, and boost its accuracy in the near-OOD setup.

In spite of all these efforts, the issue of nearly anomalous samples has not been studied in the context of ND tasks (i.e., the unsupervised setting). Furthermore, the solutions to the near-OOD problem are not directly extendable to the near-ND, as the sub-class information of the normal data is not available in the ND setup (37; 40). Furthermore, the challenge in the case of ND is that in most cases, the normal data constitutes less conceptual diversity compared with the OOD detection setup, making the uncertainty estimation challenging, especially for the nearly abnormal inputs. One has to note that some explicit or implicit form of uncertainty estimation is required for ND and out-of-distribution detection. This makes near-ND an even more difficult task than near-OOD detection.

Apart from these obstacles in extending near-OOD solutions to the near-ND problem, we note that elements of these solutions, which are outlier exposure, and generating anomalous samples adaptively through GANs are both less effective for near-ND. It is well known that the performance of outlier exposure (OE) techniques significantly depends on the diversity and distribution shift of the outlier dataset that is used for the training. This makes it difficult for OE to be used in the domains such as medical imaging, where it is hard to find accessible real outliers. In addition, unfortunately, most GAN models suffer from (1) instability in the training phase, (2) poor performance on high-resolution images, and (3) low diversity of generated samples in the context of ND (39). These challenges have prevented their effective use in ND. To address the mentioned challenges, we propose to use a "non-adversarial" diffusion-based anomaly data generation method, which can estimate the true normal distribution accurately and smoothly over the training time. Here, our contribution is to shed light on the capabilities of the recently proposed diffusion models (43), in making near-distribution synthetic anomalies to be leveraged in the training of ND models. By providing comprehensive experiments and visualizations, we show that a prematurely trained SDE-based model can generate *diverse*, *high quality*, and *non-noisy* near-outliers, which considerably beat samples that are generated by GANs or obtained from the available datasets for tuning the novelty detection task. The importance of artifact- and noise-free anomalous samples in fine tuning is due to the fact that deep models tend to learn such artifacts as shortcuts, preventing them from generalization to the true essence of the anomalous samples. Finally, our last contribution is to show that fine-tuning simple baseline ND methods with the generated samples to distinguish them from the normal data leads to a performance boost for both ND and near-ND. We use nine benchmark datasets that span a wide variety of applications and anomaly granularity. Our method achieves state-of-the-art results in the ND setting, and is especially effective for the near-ND setting, where we improve over existing work by a large margin of up to 8% in AUROC.

## 2 PROPOSED NEAR-NOVELTY DETECTION METHOD

We introduce a two-step training approach, which even can be employed to boost the performance of most of the existing state-of-the-art (SOTA) models. Following the current trend in the field, we start with a pre-trained feature extractor as (41; 4; 34) have shown their effectiveness. We use a ViT (8) backbone since (9) has demonstrated its superiority on the near out-of-distribution detection. In the first step, a fake dataset of anomalies is generated by a SDE-based diffusion model. We quantitatively and qualitatively show that the generated fake outliers are high-quality, diverse, and yet show semantic differences compared to the normal inputs. In the second step, the pre-trained backbone is fine-tuned by the generated dataset and given normal training samples through optimizing a binary classification loss. Finally, all the normal training samples are passed to the fine-tuned feature extractor, and their embeddings are stored in a memory, which is further used to obtain the $k$-NN distance of each test

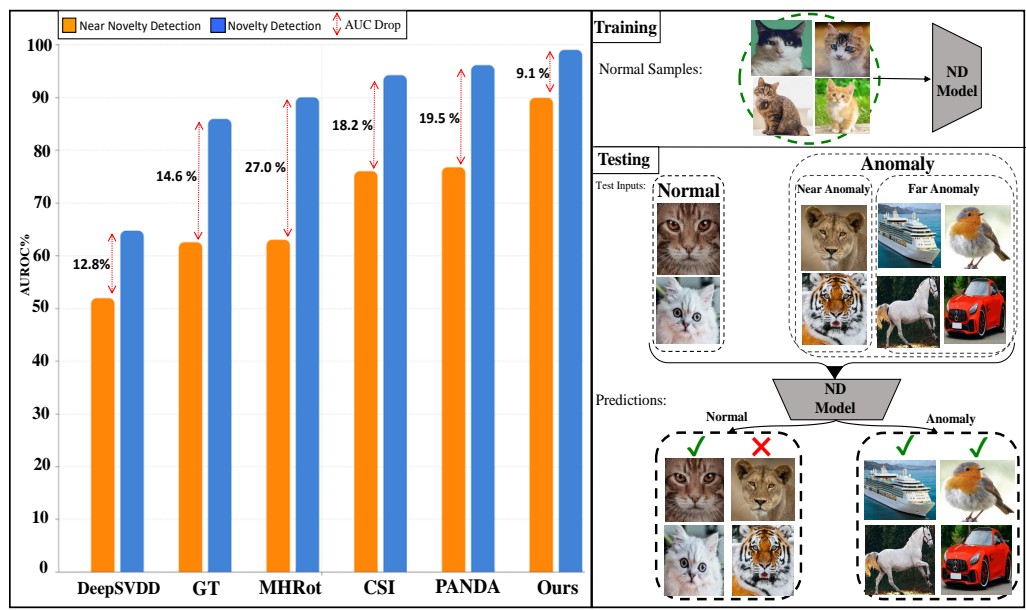

Figure 1: Sensitivity of state-of-the-art anomaly detectors when differences between normal and anomalous samples are subtle. For each model, its performance in detecting far (Blue) and near (Orange) outliers has been reported. A single class of CIFAR-10 is considered as normal, the rest of the classes are considered far anomalies, and the semantically nearest class of CIFAR-100 to the normal class is the source of near anomalies. This is the class that corresponds to the lowest test AUROC in anomaly detection. A considerable performance drop happens for the most recent methods, which means the learned normal boundaries are not tight enough and need to be adapted. For instance, while some might be able to distinguish a cat image from a car, they are still unable to detect tiger or lion images as anomalies.

input sample. This distance could serve as an anomaly score. The method is summarized visually in Fig. 2.

## 2.1 Proposed Pipeline

**Step 1 : Near Anomaly Generation** In order to generate near anomalous samples, a basic diffusion model, called Stochastic Differential Equation (SDE) (43) is adopted. In the diffusion models, the gradient of the logarithmic probability density of the data with respect to the input is called the score. A SDE model considers a continuum of distributions that develop over time in accordance with a diffusion process, to corrupt a sample and turn it into pure noise. This is called the forward process. The model is trained to learn how to gradually transform the noisy image back to its original form through the score fucntion (backward process). Running the backward process allows us to seamlessly transform a random noise into the data that could be used for sample generation as follows:

$$x_{n-1} - x_n = [f(x,t) - g^2(t)s_\theta(x,t)]dt + g(t)d\hat{w}, \quad x_T \sim N(0,1), \tag{1}$$

where $x_n$ is the data sample at time $n$, $f = 0$ and $g = \sigma^t$ are called the drift and diffusion coefficients, respectively. $s_\theta(x,t)$ is called the score function, as is expected to estimate $\nabla_x \log p_t(x)$. $\hat{w}$ represents a Brownian motion. The score function $s_\theta(x,t)$ is usually modeled by a U-Net, whose weights are $\theta$, in case $x$ is an image. The loss function to train this network is typically set to enforce $s$ to estimate $\nabla_x \log p_t(x)$:

$$\min_\theta \mathbb{E}_{t \sim U(0,T)} \mathbb{E}_{x_0 \sim D} \mathbb{E}_{x_t | x_0} \lambda(t) \| s_\theta(x_t, t) - \nabla_{x_t} \log p_t(x_t | x_0) \|_2^2, \tag{2}$$

where $\lambda(t)$ is a positive weighting function to account for change in scale of the loss across different values of $t$. One has to note that the conditional likelihood $p_t(x_t | x_0)$ takes the form of a Gaussian distribution, as a result of accumulation of normal noises over time in the forward process. For more details, see (43).

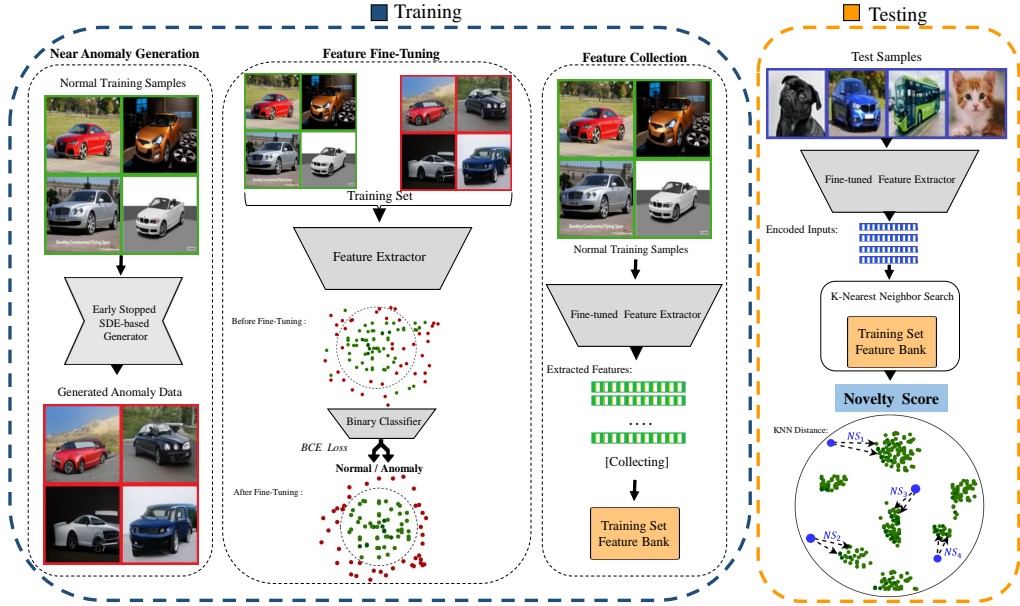

Figure 2: Overview of our framework for near distribution novelty detection. In the first step, the SDE diffusion model generates semantically close outliers, after its training is early-stopped. As it is shown, they have very subtle yet semantic differences with the normal data distribution (down-left). After that, using a linear layer, a pre-trained feature extractor is fine-tuned to solve a binary classification between the normal and abnormal inputs. This modifies the normal boundaries toward including more distinctive features. Finally, normal embeddings are stored and used to compute the $k$-NN distance at the test time as the novelty score.

The main benefit of using a score-based generative model is that in the backward process (Eq. 1), the score function denoises the input gradually that results in a relatively smooth decline in the Fréchet Inception Distance (FID) (17) across training epochs. This enables us to stably produce near anomalous samples based on the FID score of the generated outputs, which could be achieved by stopping the training process earlier than the stage where the model achieves its maximum performance. This is empirically assessed in Fig. 3 left. Note that a totally different training trajectory is obtained in GANs, where the FID oscillates and premature training does not necessarily produce high-quality anomalous samples. That is why methods like OpenGAN rely on a validation outlier dataset to determine where to stop. In addition, Fig. 3 right, shows that the quality and closeness of the samples that are generated by the diffusion model are visually more appealing that the SOTA GAN-based models.

**Step 2 : Feature Fine-tuning and Collection** Having generated a high-quality fake dataset, a lightweight projection head is added to the feature extractor, and the whole network is trained to solve a binary classification task between the given normal and generated abnormal inputs. This way, normal class boundaries are tightened and adjusted according to the abnormal inputs. After the training, all the normal embeddings are stored in the memory $\mathcal{M}$ that is used at the test time for assigning abnormality score to a given input. Note that due to the feature fine-tuning phase of our approach, the borders of the normal samples become well-structured, thus better representing the normal semantic features.

**Step 3 : Testing Phase** At the test time, for each input $x$, its $k$ nearest neighbours are found in the $\mathcal{M}$, which are denoted by $m_x^1, m_x^2,..., m_x^k$. Finally, the novelty score is found as follows:

$$\text{Novelty Score}(x) = \sum_{n=1}^{k} \|x - m_x^n\|^2 \tag{3}$$

## 2.2 DIFFUSION MODELS VS. GANS

Here, we provide some rationale on the preference of diffusion models over GANs for the the task of near-ND. Although diffusion models showed great promise in image-based generative models in recent years, some flavors of GANs, like StyleGAN-XL (42), still outperform diffusion models even in small datasets such as CIFAR-10. Therefore, it is not immediately clear, which class of generative models should be applied for the near-ND task.

We note that in diffusion models, such as DDPM, the model is trained by optimizing the Evidence Lower Bound (ELBO) (18). That is, the objective function is to minimize an upper bound on the negative evidence:

$$\mathbb{E}_{x \sim q}[- \log p_\theta(x)], \tag{4}$$

where $q$ and $p_\theta$ are the original and estimated distribution of the data, respectively. Here, $\theta$ stands for the denoiser parameters in the DDPM. But, we note that maximizing the evidence is equivalent with minimizing the Kullback-Leibler (KL) divergence between $q$ and $p_\theta$:

$$\mathbb{E}_{x \sim q}[- \log p_\theta(x)] = \mathbb{E}_{x \sim q}\left[- \log\left(\frac{p_\theta(x)}{q(x)}\right)\right] + H(q) = D(q \parallel p_\theta) + C \tag{5}$$

Therefore, if the gap in the ELBO bound is small, minimizing the DDPM loss is equivalent with making the estimated generative distribution $p_\theta$ close to the original data distribution $q$. Hence, we expect the estimated generative distribution $p_\theta$ ends up *sufficiently* close to $q$ if the training is prematurely stopped, and therefore generates "near-distribution" patterns. It should be noted that variational autoencoders (VAEs) also follow the same logic, but because of their limited capacity, which stems from their low dimensional latent space, fail to make the KL-divergence $D(q \parallel p_\theta)$ small enough (see Table 3 for the comparisons). One may also wonder why GANs do not exhibit the

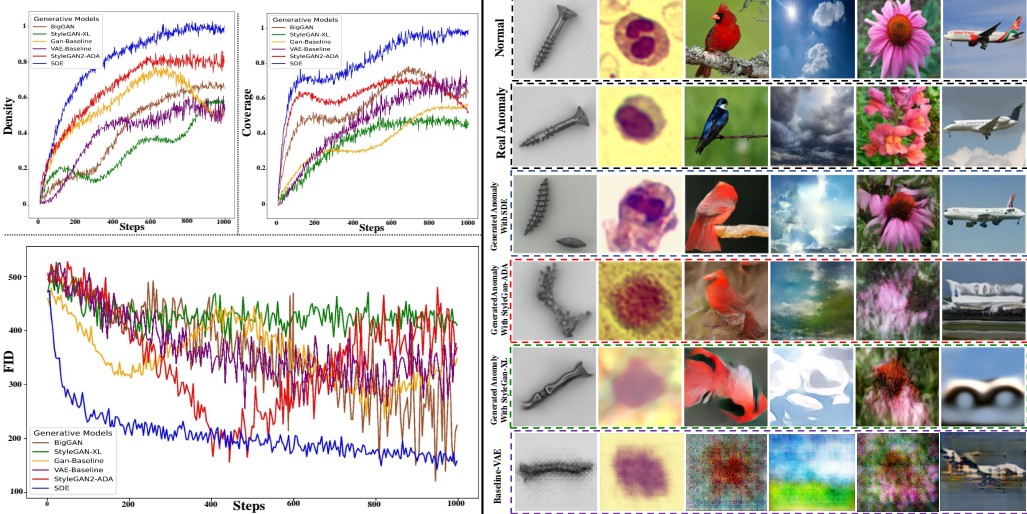

Figure 3: Left: Unlike GAN-based models, the diffusion SDE model performs much better in consistently improving the fidelity and diversity of the generated samples along the training. The metrics "density" and "coverage" (27) are used to reliably measure these concepts. Thus, obtaining high-quality and diverse anomalous samples is feasible by early stopping in the SDE model. Right: Some generated samples by various methods in different datasets as well as their corresponding normal samples are shown. As can be seen SDE samples, in the third row, are authentic (differ from the normal class), high-quality, and have minimal differences to the normal class, while other methods produce low quality samples that are far away from the normal distribution.

same property. It is also well-known that if one uses an optimal discriminator for a given generator, the generator loss becomes proportional to the Jensen-Shannon divergence of the original and estimated distributions (13). However, this does not happen in GANs, due to practical considerations, as the discriminator and generator are alternatively updated through SGD. Hence the discriminator is not necessarily optimal in each step. Therefore, early stopping does not necessarily lead to near-distribution generators. Our empirical results also confirm that various quality metrics, such as coverage and density(28), which were previously proposed to evaluate synthetic images' fidelity and diversity, where larger values that are close to 1.0 indicate better fidelity and diversity, respectively. The detailed information can be found in the Appendix 7.1. Theses metrics decline or fluctuate significantly for GANs along the training. However, such metrics exhibit a steady and consistent increase till the estimated distribution $p_\theta$ gets sufficiently close to the true distribution $q$ (see Fig. 3 left).

## 3  EXPERIMENTAL RESULTS

In this section, we compare our method against the SOTA in the ND and near-ND setups. All the experiments are performed 10 times and the average result is reported. Training details and the full dataset descriptions are provided in the Appendix 7. We compare our approach with current top self-supervised and pre-trained feature adaptation methods in the literature (44; 34; 33; 7; 16). Results already reported in the original papers were copied. Also, if the results were not reported in the original papers, we ran the experiments whenever possible.

Our experiments are organized into three categories and are presented separately in sub-tables. The following subsections will provide an overview of each category.

Additionally, we considered novelty detection in multi-class unlabeled settings, which you can see in Appendix 9.3.

### 3.1  REGULAR ND BENCHMARK

Following previous works, we evaluate the methods on the standard ND datasets. According to the standard ND protocol, multi-class datasets are first converted into an ND task by setting a class as normal and all other classes as anomalies. This is performed for all classes, in practice turning a single dataset with $C$ classes into $C$ datasets (41; 33; 34). The datasets that are tested in this setting include CIFAR-10 (23), CIFAR-100 (23), MVTecAD (3), Birds (46), Flowers (30). Table 1(a) provides the results of the case, which indicates our superiority.

### 3.2  FINE-GRAINED DATASETS FOR NEAR-ND BENCHMARK

To evaluate the performance on near-ND benchmark, we use 4 different datasets that naturally show slight differences between their class distributions. These datasets are not commonly used in ND but used in fine-grained image classification or segmentation, containing hard-to-distinguish object classes. Thus they are suitable for benchmarking near-ND. The datasets that are tested in this setting include FGVC-Aircraft (25), Stanford-Cars (22), White Blood Cell (WBC) (49), and Weather (10). Table 1(b) provides the results in this category, which again shows that our method outperforms SOTA by roughly 8 percentage points of AUROC in the mentioned near-ND setup.

### 3.3  PROPOSED DATASETS FOR NEAR-ND BENCHMARK

To further show the effectiveness of our approach, we propose the evaluation procedure represented in the Table 1(c). This setup includes CIFAR-10-FSDE, CIFAR-10-FSGAN, and CIFAR-10vs100, which are described in more details below.

#### 3.3.1  CIFAR-10vs100 BENCHMARK

In this setup, each class of CIFAR-10 ($C_i^{10}$) and its closest class in CIFAR-100 ($C_j^{100}$) are selected, to represent normal and abnormal classes, respectively, where $i$ and $j$ are class numbers that are less than 10 and 100, respectively. Then, the model is trained on the training samples of ($C_i^{10}$) and

tested against the aggregation of $(C_j^{100})$ and $(C_i^{10})$ test sets. As it is shown, almost all the methods considerably lag behind our method by a large margin.

### 3.3.2 CIFAR-10-FSDE BENCHMARK.

To show the quality of the proposed generated dataset, we evaluate the performance of some of the most outstanding ND methods when the generated dataset is considered as anomalous at the test time. We call the dataset CIFAR-10-FSDE. In this setting, one class of CIFAR-10 is selected as normal and the test set is made using the corresponding test samples of the normal class and a random subset of the fake dataset, as abnormal samples, with the same size as the normal class.

### 3.3.3 CIFAR-10-FSGAN BENCHMARK.

The anomalous data that are generated by StyleGAN-ADA contain lots of artifact-full and distorted samples that a practical novelty detector is expected to detect. Table 1 (c) compares our method with other SOTA methods in detecting such samples. Due to learning more distinguishing features, we pass the SOTA by 12 percentage points in the AUROC score, showing the applicability of our approach in detecting a wide range of anomalies.

Table 1: The performance of novelty detection methods (AUROC %) in near-ND and ND on various datasets. Our method outperforms SOTA by significant margins on all settings. Note that except CSI, all the other SOTA approaches employ a pre-trained architecture, particularly Transformaly (7), that utilizes a ViT-backbone.

(a) Comparison of AUROC of SOTA to our proposed method in the regular ND setting.

| Datasets | From Scratch | Pre-trained | | | | | |
|---|---|---|---|---|---|---|---|
| | CSI (ResNet-18) | PatchCore (WideResNet50) | PANDA (ResNet-152) | MSAD (ResNet-152) | MSAD* (ViT-B_16) | Transformaly (ViT-B_16) | Ours (ViT-B_16) |
| Birds | 52.4 | 58.1 | 95.3 | 96.7 | 93.3 | 97.8 | **98.5** |
| CIFAR-10 | 94.3 | 67.2 | 96.2 | 97.2 | 94.1 | 98.3 | **99.1** |
| CIFAR-100 | 89.6 | 64.1 | 94.1 | 96.4 | 93.0 | 97.3 | **98.1** |
| Flowers | 60.8 | 74.8 | 94.1 | 96.5 | 98.6 | **99.9** | **99.9** |
| MvTecAD | 63.6 | **99.6** | 86.5 | 87.2 | 85.5 | 87.9 | 86.4 |

\* For greater comparability, MSAD was also reimplemented with a VIT-backbone.

(b) Comparison of AUROC of SOTA to our proposed method in the natural near-ND setting.

| Datasets | From Scratch | Pre-trained | | | | |
|---|---|---|---|---|---|---|
| | CSI (ResNet-18) | PANDA (ResNet-152) | MSAD (ResNet-152) | MSAD* (ViT-B_16) | Transformaly (ViT-B_16) | Ours (ViT-B_16) |
| FGVC-Aircraft | 64.6 | 77.7 | 79.8 | 81.3 | 84.0 | **88.7** |
| Stanford-Cars | 66.5 | 87.6 | 87.1 | 85.7 | 86.7 | **90.8** |
| WBC | 50.4 | 87.4 | 87.0 | 83.6 | 85.1 | **91.2** |
| Weather | 91.5 | 81.5 | 92.4 | 85.9 | 94.3 | **97.0** |

\* For greater comparability, MSAD was also reimplemented with a VIT-backbone.

(c) The performance of the ND methods compared to ours in AUROC % in the proposed near-ND setting.

| Setting | Dataset | From Scratch | | | | | Pre-trained | | | | | |
|---|---|---|---|---|---|---|---|---|---|---|---|---|
| | | DeepSAD* | GT | MHRot | CSI | One-class-OpenGAN† | ADIB* | DN2 | PANDA | MSAD | Transformaly | Ours |
| | CIFAR-10vs100 | 50.0 | 62.6 | 63.1 | 76.1 | 50.0 | 60.7 | 58.5 | 76.8 | 79.5 | 82.3 | **90.0** |
| near-ND | CIFAR-10-FSDE | 53.5 | 60.5 | 64.5 | 87.4 | 58.1 | 52.0 | 61.7 | 57.7 | 64.9 | 75.0 | **96.4** |
| | CIFAR-10-FSGAN | 61.8 | 70.4 | 72.8 | 83.1 | 64.4 | 55.4 | 71.3 | 66.1 | 69.4 | 82.2 | **95.1** |

\* Require Extra Outlier Exposure Datasets.
† One-class-OpenGAN represents the OpenGAN adapted for our setting.

While Transformaly, PANDA, and CSI achieve very close results in the ND setup (see Table 1 (a)), they work significantly worse in the near-ND setting. To be specific, most of the methods significantly fail to detect generated fake samples, which is particularly the case for PANDA (34) with 40 percentage points performance drop. Surprisingly, DeepSAD with roughly 30 percentage points performance decrease does not work decently despite being exposed to real outliers during

the training process. To further validate our method, we adapt OpenGAN, as SOTA OOD detection, to the ND setting and report its near-ND results. Surprisingly, even by employing fake abnormal samples generated by a GAN and being exposed to real outliers, it is not able to achieve a decent performance in the near-ND setting (Table 1 (c)), which again supports the benefits of our proposed approach for generating near distribution abnormal samples. All of these results highlight the need of benchmarking previous ND methods in the near-ND setup.

## 4 ABLATION STUDIES

**Sensitivity to Backbone Replacements.** Table 2 shows our method sensitivity to the backbone replacements. All the backbones are obtained and used from (9). The results are provided for both the ND and near-ND evaluation setups. For each backbone, the performance with and without the feature fine-tuning phase based on the generated data is reported (denoted as "Pre-trained" and "Fine-tuned" in the table), showing a consistent improvement across different architectures. The performance boost is surprisingly large for some backbones such as ResNet-152, ViT-B_16, and R50+ViT-B_16 with roughly 16%, 25%, and 36% boosts in AUROCs in the near-ND setting. Moreover, ViT-B_16 and R50+ViT-B_16 perform roughly 8% and 16% better after the fine-tuning in the ND setup, showing the *generality* of our approach regardless of the setting or backbone.

Table 2: The effectiveness of our approach over different backbones (in AUROC %) in the ND and near-ND settings. The results show consistent improvements regardless of the backbone or setting that is used.

| Setting | Dataset | Training | Models | | | | |
|---|---|---|---|---|---|---|---|
| | | | ResNet-152 | ViT-B_16[*] | ViT-B_32[†] | R50+ViT-B_16[*] | ConvNeXt-B[†] |
| ND | CIFAR-10 | Pre-trained | 92.5 | 91.0 | 95.3 | 82.2 | 97.1 |
| | | Fine-tuned | 95.3 | **99.1** | 98.3 | 98.8 | 97.8 |
| near-ND | CIFAR-10vs100 | Pre-trained | 58.5 | 65.5 | 66.3 | 50.0 | 71.5 |
| | | Fine-tuned | 74.7 | **90.0** | 78.9 | 85.9 | 81.1 |

[*] Pretrained on ImageNet 21K.
[†] Pretrained on ImageNet 21K and fine-tuned on ImageNet 1K.

**SDE vs. Others.** In this experiment, the effect of using different generative models is explored. All the experiments are done in the one-vs-all setting. Table 3 shows clear superiority of using a SDE-based model compared to the well-known generative models (20; 42; 5; 14) with 2% to 20% better AUROCs based on the chosen dataset. Evidently, employing our fake samples generated by the SDE is almost always beneficial as opposed the ones generated by the other models. Interestingly, other models generated samples are harmful to the performance in Weather and MvTecAD datasets. Having artifacts, mode-collapse, and undiversified generated samples could be the reasons behind this observation, which are shown in Appendix10. This highlights that not all the generative models are beneficial for the ND and near-ND tasks, and they need to be carefully selected based on the constraints defined in such domains.

Table 3: Comparison of our model performance (in AUROC %) upon using different generative models for each class of the CIFAR-10, Weather, WBC and MvTecAD dataset. The performance of the base backbone is also included to measure the effectiveness of the training process. Diffusion-based data generation considerably passes the GAN-based SOTAs in the novelty detection task. Coverage&Density metrics are calculated according to the dataset and generative model. These metrics measure the diversity and fidelity of the samples that are generated by generative models.

| Dataset | Metric | Base Backbone | Generative Model | | | | | | |
|---|---|---|---|---|---|---|---|---|---|
| | | (without fake data) | Baseline-VAE | Baseline-Gan | BigGAN | StyleGAN-ADA | StyleGAN-XL | DenseFlow | SDE |
| CIFAR10 | AUROC | 91.0 | 93.8 | 94.5 | 96.9 | 97.8 | 96.5 | 96.4 | **99.1** |
| | Density&Coverage | - | (0.537/0.473) | (0.660/0.581) | (0.732/0.769) | (0.814/0.885) | (0.786/0.638) | (0.704/0.635) | **(0.872/0.944)** |
| Weather | AUROC | 86.5 | 69.6 | 83.7 | 85.3 | 82.4 | 76.8 | 84.9 | **97.0** |
| | Density&Coverage | - | (0.583/0.463) | (0.642/0.681) | (0.640/0.438) | **(0.782/0.765)** | (0.658/0.514) | (0.628/0.741) | (0.737/**0.796**) |
| WBC | AUROC | 83.1 | 75.2 | 80.4 | 84.6 | 85.3 | 77.2 | 82.0 | **91.2** |
| | Density&Coverage | - | (0.607/0.384) | (0.719/0.483) | (0.535/0.684) | (0.810/0.761) | (0.632/0.578) | (0.754/0.827) | **(0.856/0.873)** |
| MvTecAD | AUROC | 82.5 | 68.8 | 71.8 | 76.4 | 80.5 | 73.2 | 78.3 | **86.4** |
| | Density&Coverage | - | (0.598/0.402) | (0.641/0.678) | (0.684/0.587) | (0.826/0.641) | (0.428/0.364) | (0.761/0.526) | **(0.908/0.835)** |

**Sensitivity to $k$-NN and Stopping Point.** Table 4 shows the performance stability with respect to the number of nearest neighbours for both the ND and near-ND setups. Clearly, the method is barely sensitive to this parameter. DN2 (2) has provided extensive experiments on the effectiveness of applying $k$-NN to the pre-trained features for the AD task, claiming $k = 2$ is the best choice. We also observe that the same trend happens in our experiments. Similarly, the method is robust against the stopping point with at most 2% variation for reasonable FID scores, making it practical for the real-world applications.

Table 4: The sensitivity of the proposed method to the $k$-NN parameter and training stopping point based on the FID score.

| Setting | Dataset | KNN | | | | | Stopping Point | | | |
|---------|---------|-----|-----|-----|------|------|--------|-------------|----------|--------|
| | | k=1 | k=2 | k=5 | k=10 | k=50 | 300<FID | 100<FID<200 | 30<FID<50 | FID<20 |
| ND | CIFAR-10 | 99.0 | 99.1 | 98.9 | 98.9 | 98.7 | 96.8 | 97.7 | 99.1 | 92.6 |
| near-ND | CIFAR-10vs100 | 89.8 | 90.0 | 90.0 | 90.0 | 89.7 | 82.7 | 87.1 | 90.0 | 68.2 |

## 5 RELATED WORK

**Outlier Exposure Based Approaches** Utilizing the fake data for the novelty detection task has previously been considered. The general idea is to employ synthetic images, which may be generated by GANs, to augment the training set (6; 26; 50; 29; 11; 1). In the case of open-set recognition, (21) proposed OpenGAN, which adversarially generates fake open-set images. The discriminator is then utilized at the test time for the novelty detection. The key point in OpenGAN is that a tiny additional dataset, containing both in- and outliers, is used as a validation set for the model selection. This, also known as outlier exposure, is necessary due to the GAN unstable training. Even though these models can work well on simple datasets, they cannot handle more complex datasets and cannot detect hard near-distribution anomalies.

**Self-supervised learning** Many studies have shown that self-supervised methods can extract meaningful features that could be exploited in the anomaly detection tasks such as MHRot (16). GT (12) uses geometric transformations including flip, translation, and rotation to learn normal features for anomaly detection. Alternatively, puzzle solving (38), and Cut-Paste (24) are proposed in the context of anomaly detection. It has recently been shown that contrastive learning can also improve anomaly detection (44), which encourages the model to learn normal features by contrasting positive and negative samples.

**Pre-trained Methods** Several works used pre-trained networks as anomaly detectors. Intuitively, abnormal and normal data do not overlap in the feature space because many features model high-level and semantic image properties. So, in the pre-trained feature space, one may classify normal vs. anomaly. (2) used the $k$-nearest neighbors distance between the test input and training set features as an anomaly score. (48) trained a GMM on the normal sample features, which could then identify anomalous samples as low probability areas. PANDA (34) attempts to project the pre-trained features of the normal distribution to another compact feature space employing the DSVDD (35) objective function. In (7; 4; 41), the pre-trained model is a teacher and the student is trained to mimic the teacher's behavior solely on the normal samples.

## 6 CONCLUSION

In this paper, we revealed a key weakness of existing SOTA novelty detection methods, where they fail to identify anomalies that are slightly different from the normal data. We proposed to take a non-adaptive generative modeling that allows controllable generation of anomalous data. It turns out that SDEs fit this specification much better compared to other SOTA generative models such as GANs. The generated data levels up pre-trained distance-based novelty detection methods not only based on their AUROCs in the standard ND setting but also in the near-ND setting. The improvements that are made by our method are consistent across wide variety of datasets, and choices of the backbone model.

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

# Appendix

## 7 Experimental Settings

We use a ViT-B_16 as the feature extractor (pretrained on ImageNet 21k), learning rate = 4e-4, weight decay = 5e-5, batch size = 16, optimizer = SGD, and a linear head with 2 output neurons. We freeze the first six layers of the network, and the rest is fine-tuned till convergence for all the experiments. For the data generation phase, we use the SDE-based model explained in (43) with exactly the same setup default hyper-parameters e.g. we used predictor-corrector as a sampler. The inputs are resized to 224×224, and as a rule of thumb, the data generation is stopped on FID $\approx 40$ and FID $\approx 200$ for low and high-resolution datasets. All the results in our tables are either reported from the reference papers or run by us using their official repositories.

### 7.1 Density & Coverage

The density and coverage metrics can be used to assess the diversity and fidelity of generative models, respectively. The method measures the distance between the generated and real images using a manifold estimation procedure code (19). In order to exploit feature representations for these metrics, features prior to the final classification layer of a pre-trained model are used. The following is a mathematical expression for these metrics:

$$Density\left(\boldsymbol{X}_s, \boldsymbol{X}_t, F, k\right) = \frac{1}{kM} \sum_{j=1}^{M} \sum_{i=1}^{N} \mathbb{I}\left(\boldsymbol{f}_{t,j} \in B\left(\boldsymbol{f}_{s,i}, \mathrm{NN}_k\left(F\left(\boldsymbol{X}_s\right), \boldsymbol{f}_{s,i}, k\right)\right)\right), \qquad (6)$$

$$Coverage\left(\boldsymbol{X}_s, \boldsymbol{X}_t, F, k\right) = \frac{1}{N} \sum_{i=1}^{N} \mathbb{I}\left(\exists j \text{ s.t. } \boldsymbol{f}_{t,j} \in B\left(\boldsymbol{f}_{s,i}, \mathrm{NN}_k\left(F\left(\boldsymbol{X}_s\right), \boldsymbol{f}_{s,i}, k\right)\right)\right). \qquad (7)$$

where $F$ is a feature extractor, $\boldsymbol{f}$ is a collection of features from $F$,

$\boldsymbol{X}_s = \{\boldsymbol{x}_{s,1}, \ldots, \boldsymbol{x}_{s,N}\}$ denotes real images, $\boldsymbol{X}_t = \{\boldsymbol{x}_{t,1}, \ldots, \boldsymbol{x}_{t,M}\}$ denotes generated images , $B(\boldsymbol{f}, r)$ is the n-dimensional sphere in which $\boldsymbol{f} = F(\boldsymbol{x}) \in \mathbb{R}^n$ is the center and $r$ is the radius, $\mathrm{NN}_k f(F, \boldsymbol{f}, k)$ is the distance from $\boldsymbol{f}$ to the $k$-th nearest embedding in $F$, and $\mathbb{I}(\cdot)$ is a indicator function. We use the standard InceptionV3 features, which are also used to compute the FID. The measures are computed using the official code (28).

### 7.2 OOD vs. ND

The common objective of OOD detection and novelty detection is to detect anomalies. Since they share the same goal, they may be considered the same. However, in OOD detection an extra piece of information is present, which is the class information of each normal training sample. This extra information makes the techniques for the OOD detection and novelty detection quite different. For instance, one could achieve satisfactory detection rates in OOD, by simply training a model to classify normal classes, and threshold the maximum softmax probability (15). This simple remedy is unfortunately impossible for novelty detection, where the training of the base classifier is impossible due to non-existence of this extra class information. Therefore, OOD detection methods cannot simply be used to solve the novelty detection problem. In light of this, the literature has categorized these two problems as being different.

### 7.3 Analysis of Stopping Point

The stopping point of the SDE is determined based on the ideal FID, i.e., once the SDE has converged on the training data, the FID is computed. We then set a stopping point equal to a factor (5) of the ideal FID as a rule of thumb. E.g., in the CIFAR-10, SDE achieves FID $\approx 8$. Consequently, we set the stopping point at FID $\approx 40$ for low-resolution. The reason that our FID is higher than the previously

reported FID (i.e. 2.2) is that we only have one class, resulting in a smaller training set. Also, we set up an ablation study on stopping point sensitivity in table 3, indicating that the method is relatively robust against limited variations in stopping points and the results are not fragile. It is important to note that the synthesized images were not an effective anomaly in the first training procedure because they are extremely noisy. Furthermore, one has to note that current deep models are highly capable of learning artifacts from their input, instead of paying attention to the general semantics of the input. In addition, once the model converges to the real distribution, the images generated become similar to the normal distribution, which disallows using these images as anomaly samples. Therefore, it seems reasonable that performance results in the first and last phases of the training procedure would be poor. figure 1 illustrates the mentioned trend (37).

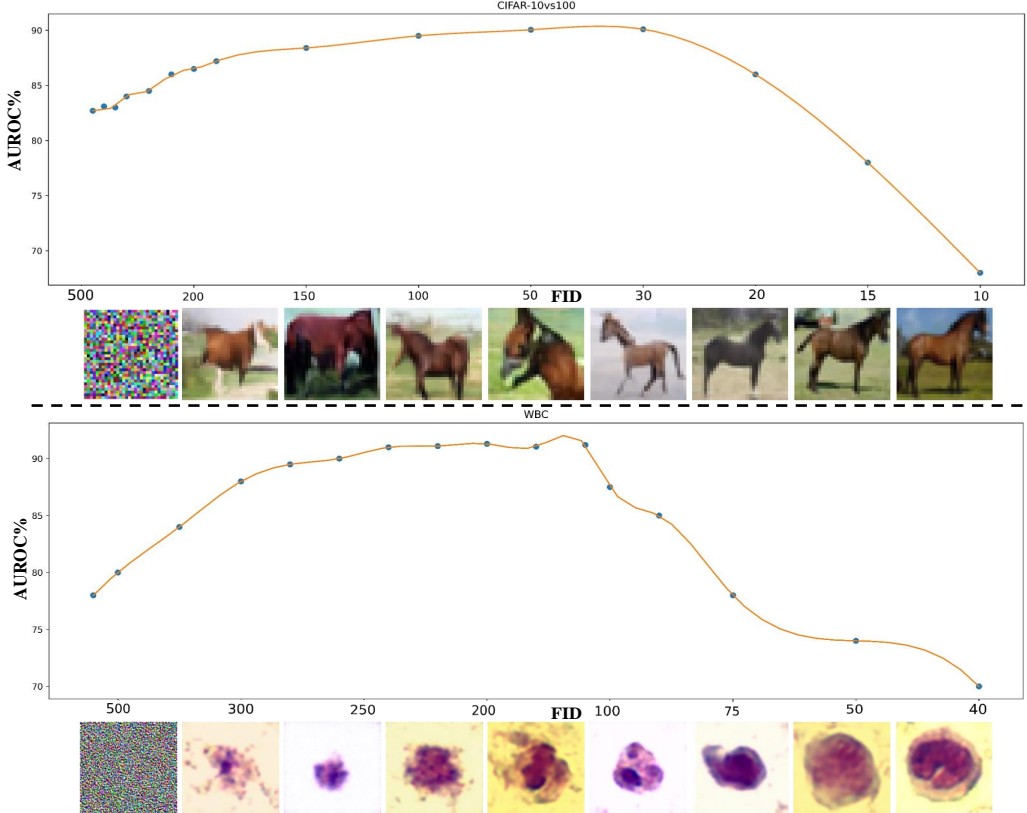

Figure 1: Spectrum of AUCs, based on Stopping points for WBC dataset and CIFAR10-vs100 setting, as proxies for low- and high-resolution data, respectively.

# 8 NEAR NOVELTY DETECTION DEFINITION

To provide a standard benchmark for the near-ND task there is a need to define a one-class distribution closeness score. Suppose a $K$-class training dataset is given from which a normal class $C_i$ is randomly sampled and used to train the model $\mathcal{M}$. The closest abnormal distribution with respect to the selected normal class for $\mathcal{M}$ can be defined as $C_j \neq C_i$ that minimizes the test time performance when it is considered as the abnormal distribution. We call the performance in the mentioned scenario the bottom-1 score of class $C_i$ with the backbone $\mathcal{M}$. However, this score depends on the choice of $\mathcal{M}$, making it specific for every problem setup and method. Therefore, inspired by the CLP criterion introduced in (47) as a measurement of dataset distance, we could use CLP as an alternative closeness score for the novelty detection task.

Given a $K$-class training dataset, one category is randomly selected as the normal distribution. Then, a supervised classifier $\mathcal{P}$ is trained on the rest $K - 1$ abnormal categories. Assume $x$ to be a training

sample of the selected normal class. The closeness score of each abnormal class $i$ with respect to the normal class is obtained as follows:

$$\text{Closeness score}_i = \sum_{x \in \text{Normal Class}} \mathcal{P}(\hat{y} = i|x).$$

(8)

The higher the closeness score of an abnormal category, the more similar it is to the normal class. The same situation also holds when abnormal categories are selected from another dataset. Fig. 1 in the Appendix indicates a decent correlation between the bottom-1 score and closeness score, implying that our proposed criterion can be considered as a proxy for the ideal score.

## 9 NEAR-ND

### 9.1 COMPARISON BETWEEN BOTTOM-1 AND CLP

In section 8, we proposed the closeness score (CLP) to find the closest abnormal class for each class of the normal dataset. In our experiments, we used the ViT-B_16 (pretrained on ImageNet-21K) as the backbone used for extracting the closest abnormal classes based on this criterion. The bottom-1 abnormal class is the class that has the lowest novelty detection performance during the testing phase. Note that the abnormal classes chosen based on the CLP criterion do not necessarily have the worst novelty detection performance; therefore, it is expected for the novelty detection methods to perform better on the CLP criterion. This section aims to investigate how well these two criteria match each other. In the Table 5, for each novelty detection model and every class of the CIFAR-10 dataset, the bottom-1 class is shown. Furthermore, the last row indicates the closest abnormal class selected based on the CLP criterion. As shown in the Table 5, the extracted classes based on the CLP criterion are the same or conceptually similar to the respected bottom-1 classes. Figure 2 also shows a decent correlation between the bottom-1 score and the CLP score, which means both these criteria could be used to extract near-distribution abnormal classes, and that the CLP could be used as a proxy to the bottom-1 criterion. Also, The Pearson correlation coefficient between these two results is 0.843, indicating a strong positive relationship.

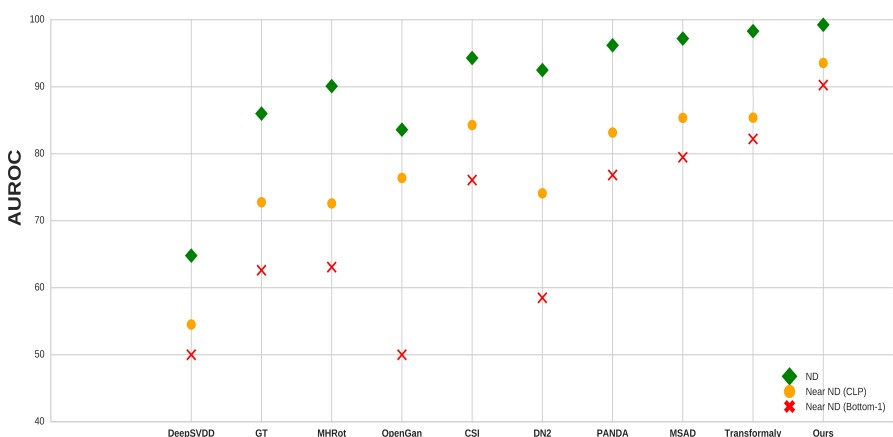

Figure 2: The performance of novelty detection methods (AUROC %) in the near-ND and ND settings. In the near-ND setting, results are reported according to both criteria, i.e. CLP and bottom-1.

### 9.2 BOTTOM-$i$ CLASSES AS THE ABNORMAL DISTRIBUTION

The Bottom-$i$ means averaging the AUROCs for the i abnormal classes that have the lowest AUROCs, e.g. bottom-100 in the case of CIFAR-100 denotes averaging the AUROC results for all 100 classes.

Table 5: For each model and every class of the CIFAR-10, the anomalous class that minimizes the anomaly detection performance among 100 classes of the CIFAR-100 dataset is reported. The last row is the CIFAR-100 class that achieves the highest CLP score. Note that the latter does not depend on the anomaly detection method.

| Method | Airplane | Automobile | Bird | Cat | Deer | Dog | Frog | Horse | Ship | Truck |
|---|---|---|---|---|---|---|---|---|---|---|
| DeepSVDD | Plain | Plain | Plain | Cloud | Plain | Mountain | Plain | Plain | Plain | Plain |
| GT | Mountain | Pickup_Truck | Camel | Fox | Elephant | Bear | Crocodile | Elephant | Train | Bus |
| MHRot | Plain | Pickup_Truck | Camel | Fox | Cattle | Fox | Crocodile | Elephant | Sea | Pickup_Truck |
| one_class OpenGan | Plain | Plain | Plain | Wardrobe | Forest | Telephone | Aquarium_Fish | Oak_Tree | Whale | Plain |
| CSI | Pickup_Truck | Pickup_Truck | Camel | Fox | Cattle | Cattle | Dinosaur | Cattle | Pickup_Truck | Pickup_Truck |
| DN2 | Plain | Pickup_Truck | Willow_Tree | Fox | Forest | Fox | Forest | Cattle | Sea | Pickup_Truck |
| PANDA | Dolphin | Pickup_Truck | Kangaroo | Fox | Kangaroo | Raccoon | Crocodile | Cattle | Bridge | Pickup_Truck |
| MSAD | Cloud | Pickup_Truck | Kangaroo | Fox | Kangaroo | Fox | Beaver | Cattle | Sea | Pickup_Truck |
| Transformaly | Cloud | Pickup_Truck | Kangaroo | Rabbit | Kangaroo | Wolf | Lizard | Cattle | Sea | Pickup_Truck |
| Ours | Tank | Pickup_Truck | Camel | Wolf | Kangaroo | Wolf | Lizard | Camel | Streetcar | Pickup_Truck |
| CLP | Rocket | Pickup_Truck | Shrew | Leopard | Cattle | Rabbit | Lizard | Cattle | Bridge | Bus |

Table 6: The performance of novelty detection methods (AUROC %) in the near-ND setting. For each method and every normal class, both results have been reported according to the CLP and bottom-1 metrics.

| Method | Metric | Airplane | Automobile | Bird | Cat | Deer | Dog | Frog | Horse | Ship | Truck | Mean |
|---|---|---|---|---|---|---|---|---|---|---|---|---|
| DeepSVDD | bottom-1 | 17.2 | 24.0 | 22.1 | 34.0 | 21.5 | 25.1 | 25.8 | 27.1 | 32.7 | 20.4 | 50.0 |
| | CLP | 52.1 | 52.3 | 53.9 | 64.7 | 53.0 | 39.9 | 57.5 | 55.6 | 61.7 | 54.5 | 54.5 |
| GT | bottom-1 | 39.2 | 61.8 | 61.0 | 54.2 | 60.1 | 65.1 | 71.0 | 70.4 | 80.5 | 62.9 | 62.6 |
| | CLP | 85.6 | 61.8 | 81.4 | 58.7 | 61.6 | 80.2 | 78.8 | 76.1 | 80.6 | 62.9 | 72.8 |
| MHRot | bottom-1 | 34.1 | 62.4 | 65.2 | 57.6 | 61.9 | 69.3 | 77.8 | 73.1 | 75.0 | 54.3 | 63.1 |
| | CLP | 77.4 | 62.4 | 81.5 | 60.4 | 61.9 | 82.1 | 82.3 | 73.3 | 79.9 | 64.7 | 72.6 |
| one_class OpenGan | bottom-1 | 11.4 | 18.1 | 10.0 | 15.3 | 19.2 | 29.8 | 16.9 | 20.9 | 40.4 | 6.2 | 50.0 |
| | CLP | 66.3 | 92.2 | 94.9 | 64.7 | 51.7 | 63.7 | 93.3 | 66.5 | 85.7 | 85.0 | 76.4 |
| CSI | bottom-1 | 73.4 | 68.4 | 81.3 | 65.0 | 71.8 | 78.1 | 88.3 | 85.9 | 91.1 | 57.5 | 76.1 |
| | CLP | 94.2 | 68.4 | 93.7 | 79.9 | 71.9 | 90.6 | 91.5 | 87.1 | 91.0 | 74.6 | 84.3 |
| DN2 | bottom-1 | 57.5 | 61.8 | 43.8 | 54.0 | 51.5 | 66.2 | 59.4 | 74.6 | 54.9 | 61.0 | 58.5 |
| | CLP | 87.5 | 61.8 | 61.7 | 66.5 | 84.4 | 83.5 | 74.3 | 74.6 | 77.0 | 69.8 | 74.1 |
| PANDA | bottom-1 | 89.8 | 70.0 | 81.7 | 50.0 | 77.0 | 74.8 | 83.8 | 80.3 | 88.7 | 72.1 | 76.8 |
| | CLP | 94.5 | 70.0 | 89.1 | 66.7 | 91.7 | 88.8 | 86.5 | 80.3 | 88.7 | 75.3 | 83.2 |
| MSAD | bottom-1 | 90.4 | 72.0 | 87.5 | 52.9 | 81.1 | 80.5 | 85.3 | 83.0 | 83.8 | 78.3 | 79.5 |
| | CLP | 94.0 | 72.0 | 89.8 | 75.3 | 87.3 | 94.5 | 90.2 | 83.0 | 88.6 | 79.0 | 85.4 |
| Transformaly | bottom-1 | 82.1 | 67.5 | 88.3 | 78.4 | 80.9 | 89.8 | 85.3 | 93.1 | 84.0 | 73.5 | 82.3 |
| | CLP | 90.1 | 67.5 | 89.0 | 79.1 | 90.1 | 93.6 | 85.3 | 93.1 | 90.2 | 74.7 | 85.3 |
| Ours | bottom-1 | 95.6 | 83.5 | 93.4 | 86.9 | 92.3 | 84.5 | 94.8 | 97.8 | 93.7 | 77.0 | **90.0** |
| | CLP | 96.3 | 83.5 | 99.2 | 89.4 | 97.0 | 95.8 | 94.8 | 98.0 | 93.8 | 79.9 | **92.8** |

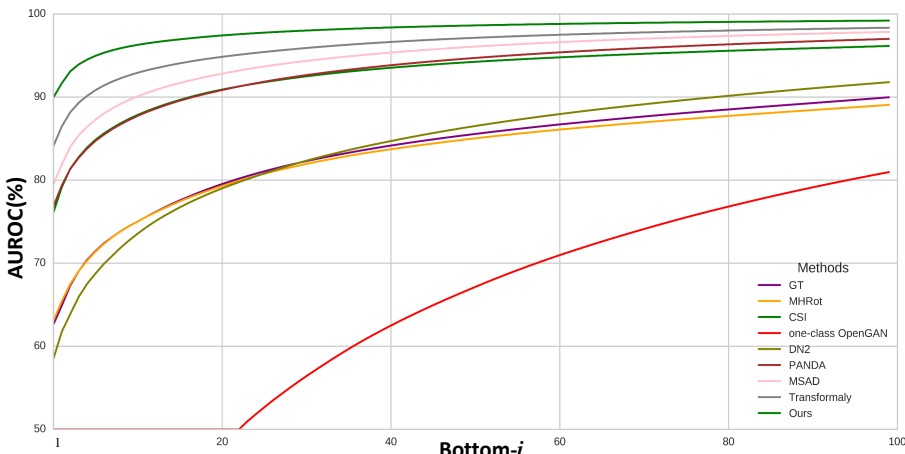

Figure 3: By increasing $i$, the AUROCs of the models improve due to the presence of more anomalies that are further from the boundary. As expected, in most cases, the gap between the performance of models has become smaller. Furthermore, in the case of $i = 1$, in which anomalies constitute the near distribution ones, the models' performances vary greatly. In this case, our proposed method achieves SOTA results by a large margin.

## 9.3 UNLABELED MULTI-CLASS

In this setup, we assume that normal samples are from a specific multi-class dataset (i.e. CIFAR-10) without labels, testing on various external datasets as abnormal samples(e.g. CIFAR-100). We copied the CSI results from the original paper.

Table 7: AUROC (%) of methods trained on unlabeled CIFAR-10

|      | SVHN | LSUN | CIFAR-100 |
|------|------|------|-----------|
| Ours | **99.9** | **99.4** | **95.7** |
| CSI  | 99.8 | 97.5 | 89.2 |

## 9.4 PER-CLASS RESULTS

In this section, we provide our model's performance for every class of the CIFAR-10 and CIFAR-100 datasets.

Table 8: Performance of our method in AUROC (%) for each class of the CIFAR-10 dataset in the one-vs-all setting.

| Method | 0 | 1 | 2 | 3 | 4 | 5 | 6 | 7 | 8 | 9 | Mean |
|--------|------|------|------|------|------|------|------|------|------|------|------|
| Ours | 99.2 | 99.4 | 99.2 | 98.1 | 99.5 | 98.1 | 99.8 | 99.5 | 99.2 | 98.8 | 99.1 |

Table 9: Performance of our method in AUROC (%) for each class of the coarse-grained CIFAR-100 dataset in the one-vs-all setting.

| Method | 0 | 1 | 2 | 3 | 4 | 5 | 6 | 7 | 8 | 9 | 10 | 11 | 12 | 13 | 14 | 15 | 16 | 17 | 18 | 19 | Mean |
|--------|------|------|------|------|------|------|------|------|------|------|------|------|------|------|------|------|------|------|------|------|------|
| Ours | 98.0 | 99.0 | 99.1 | 98.2 | 98.2 | 97.7 | 98.8 | 98.7 | 99.0 | 96.6 | 96.5 | 98.1 | 98.4 | 96.3 | 98.1 | 97.1 | 98.5 | 98.8 | 98.8 | 97.9 | 98.1 |

## 9.5 CIFAR-10 VS. CIFAR-100

In this section, we compare images of the CIFAR-10 dataset, images of respective classes selected from the CIFAR-100 dataset based on the CLP criterion, and the corresponding anomalies generated by our SDE model.

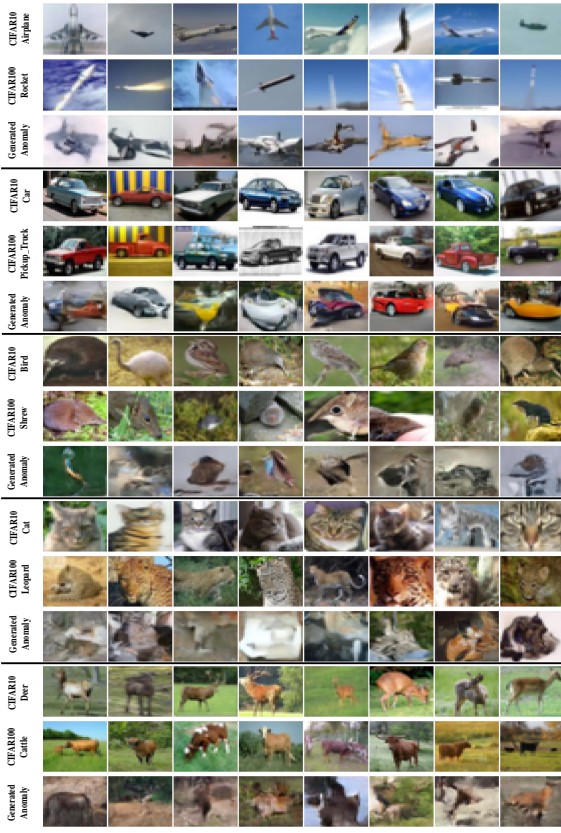

Figure 4: For each class of CIFAR-10, nearest class from CIFAR-100 has been provided, according to the closeness score. In each 3-row panel, the first row is assumed as normal, the second one is assumed the nearest abnormal class, and the third row indicates the corresponding generated fake images using the SDE model.

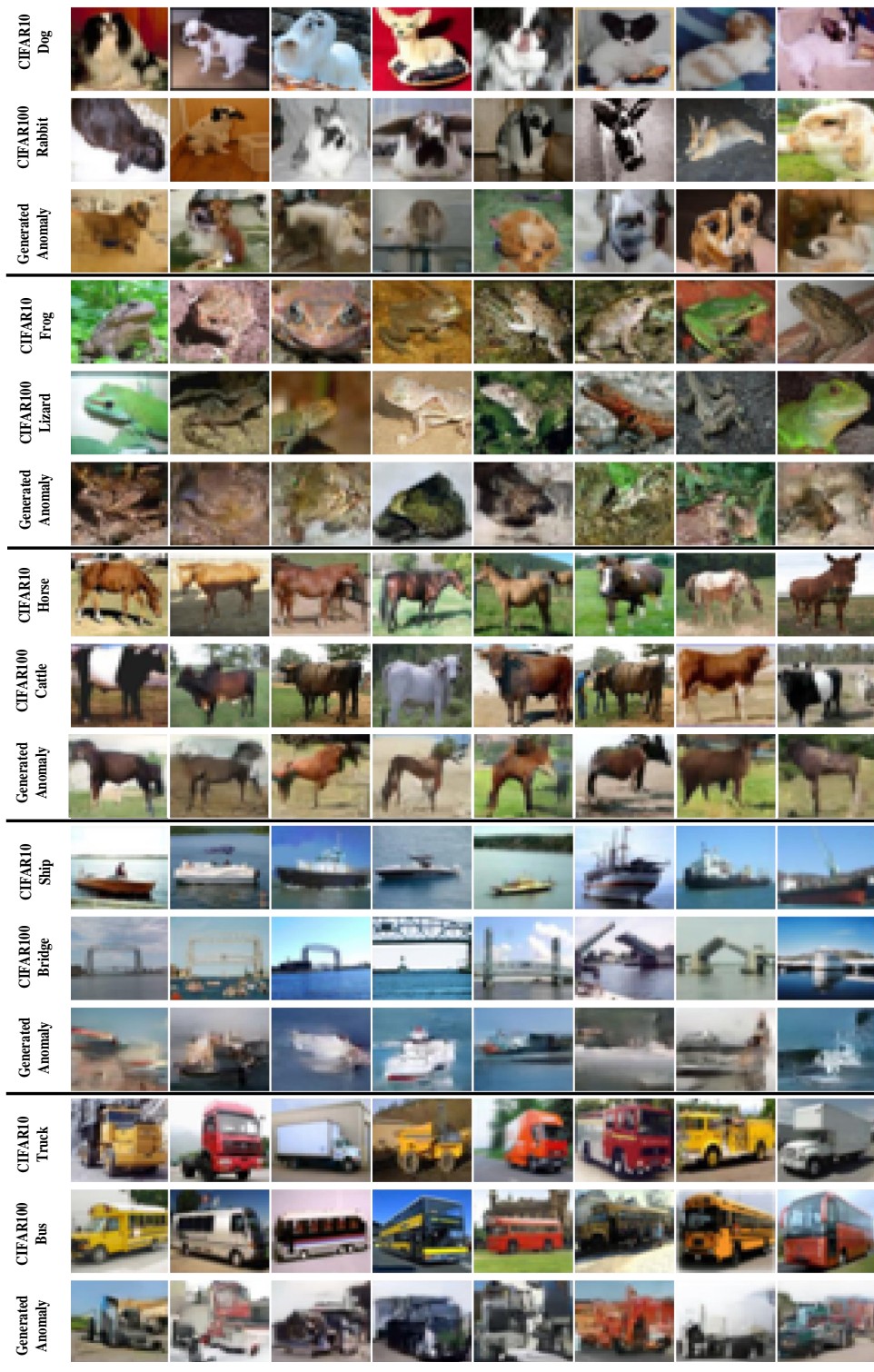

Figure 5: For each class of CIFAR-10, nearest class from CIFAR-100 has been provided, according to the closeness score. In each 3-row panel, the first row is assumed as normal, the second one is assumed the nearest abnormal class, and the third row indicates the corresponding generated fake images using the SDE model.

# 10 THE SDE-BASED GENERATIVE MODEL

## 10.1 SDE VS. OTHER GENERATIVE MODELS

In this section, according to our main setting, we have replaced the SDE with other generative models, and sampled the images generated by these models before convergence. Because in our setup, only one class (the normal class) is available for the training of the generative model. As the sample size becomes limited in such a setting, model convergence and generation of high quality real samples become a challenge. Even under such circumstances, the SDE-based models converge. Furthermore, such models can eventually generate high quality real images. However, other generative models do not converge properly, and the images that are sampled using the early stopped model are often noisy and contain artifacts. As opposed to other generative methods, the proposed premature training of the SDE yields authentic, near-distribution, diverse, and artifact-free images. This makes it a better choice for the near-ND setup.

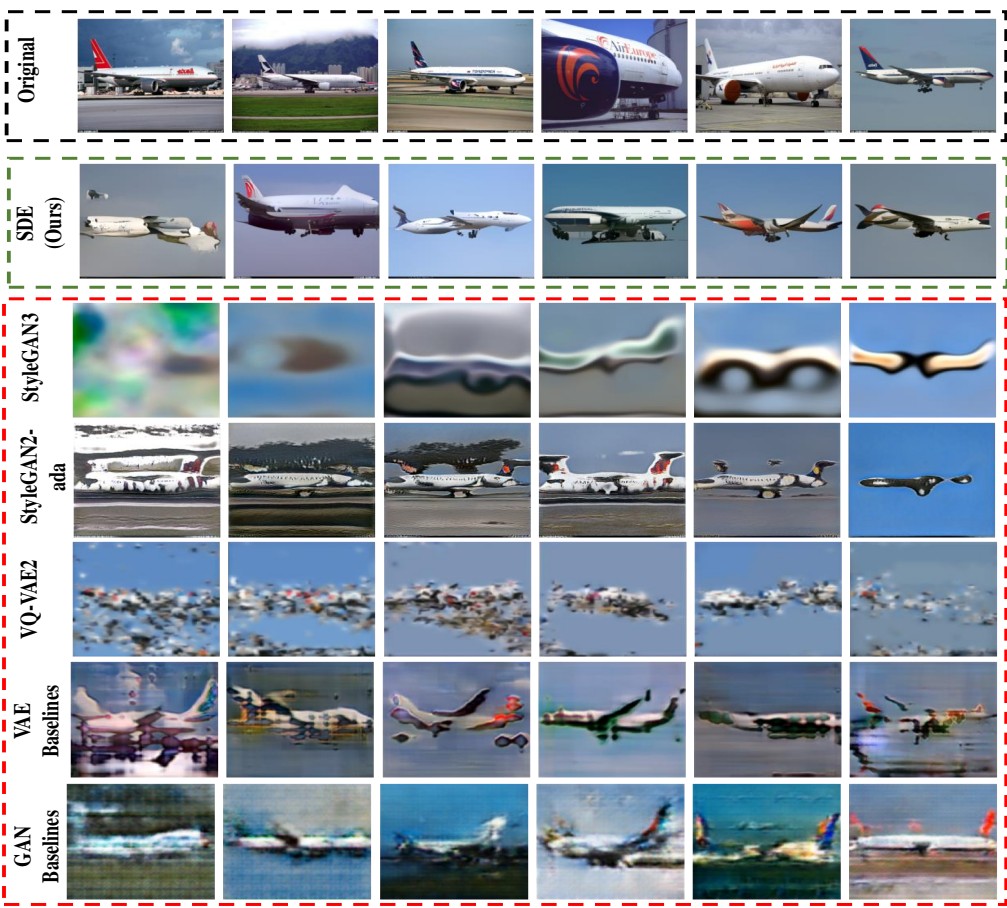

Figure 6: Generated anomalous samples by different generative models on the FGVC-Aircraft dataset.

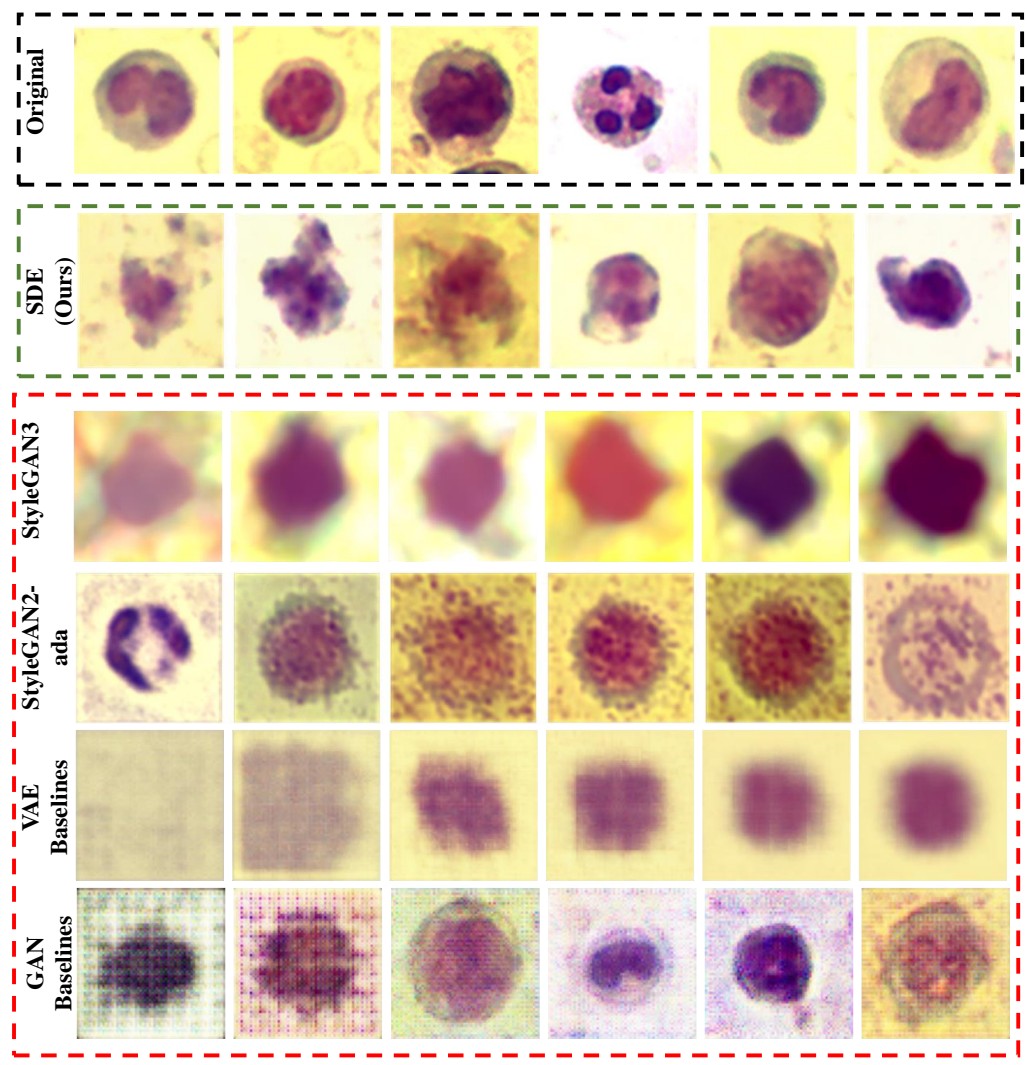

Figure 7: Generated anomalous samples by different generative models on the WBC dataset.

## 10.2 CIFAR-10-FSDE

These images are randomly plotted for each of the classes in the proposed dataset, CIFAR-10-FSDE, generated through an early stopped SDE. As can be seen, the images are clearly different semantically from the normal class. Despite this semantic difference, most novelty detection models have a poor performance in the detection of these images as anomalous.

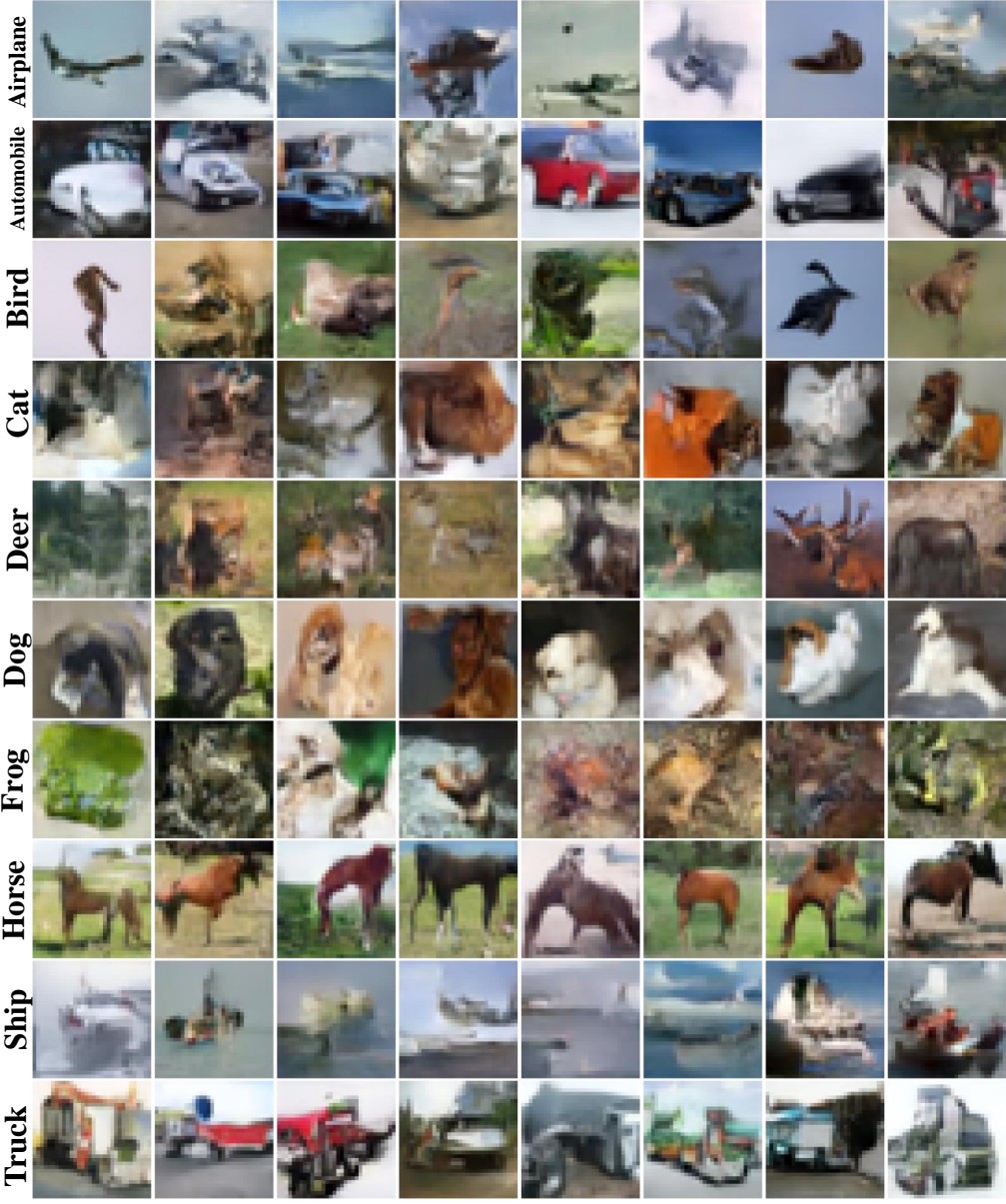

Figure 8: An overview images in the synthetic CIFAR10-FSDE dataset.

Each row is a class on which the SDE model is trained. The SDE model is trained on higher number of iterations in moving from the left to the right columns. The rightmost column contains a real images.

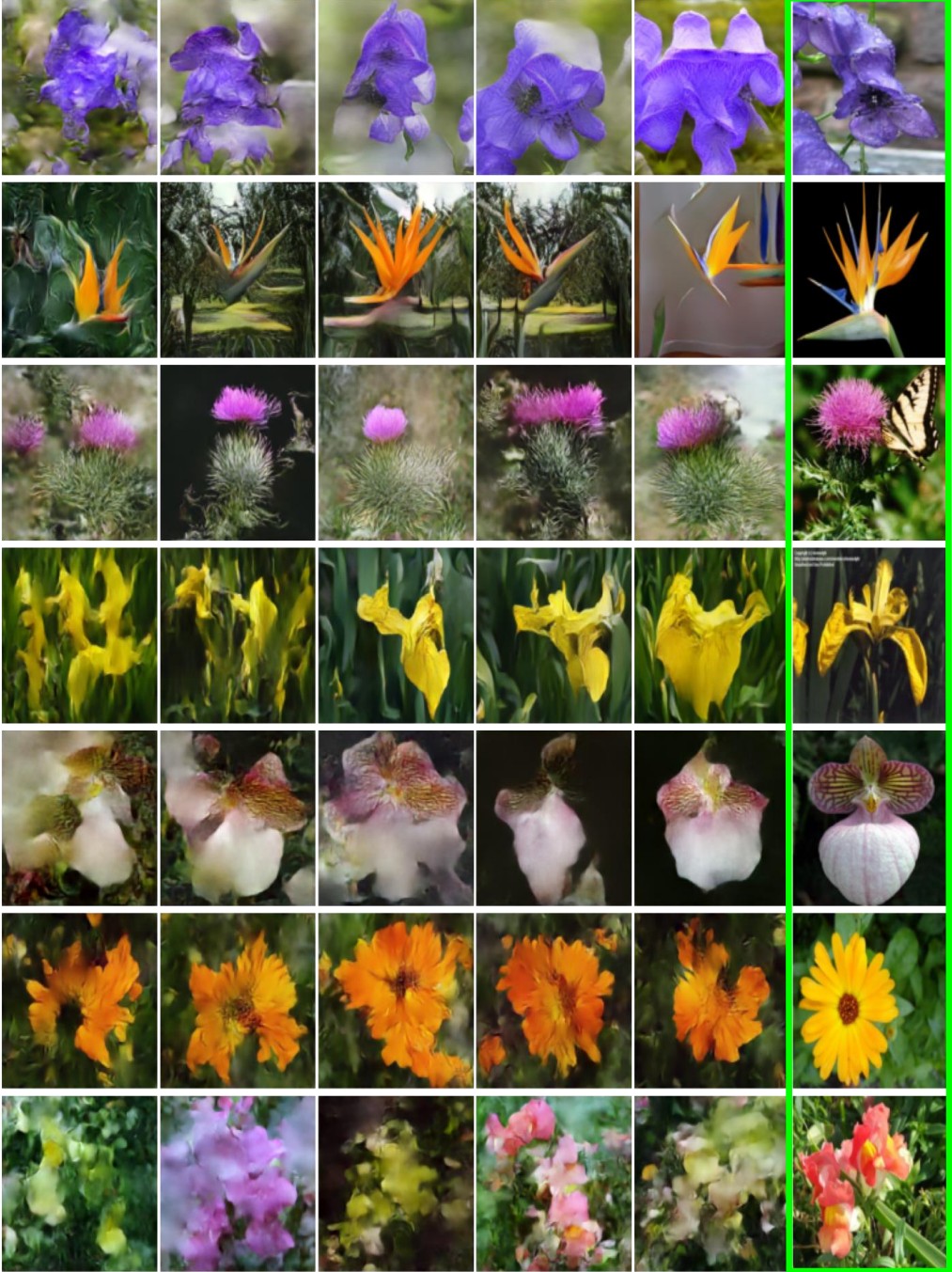

Figure 9: Generated anomaly images on 102 category of the Flowers dataset. The images highlighted in green are normal samples.

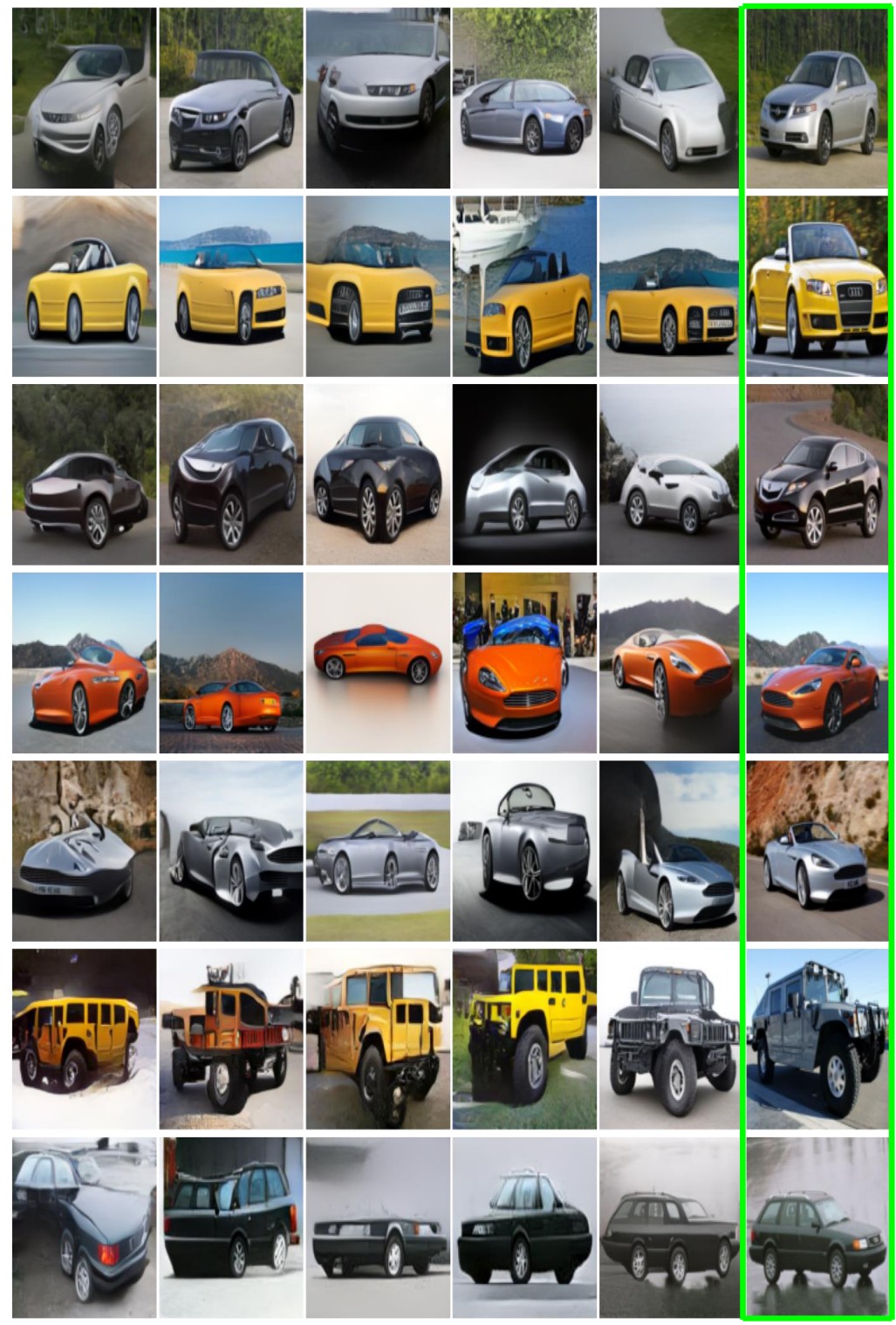

Figure 10: Generated anomaly images on the StanfordCars dataset. The images highlighted in green are normal samples.

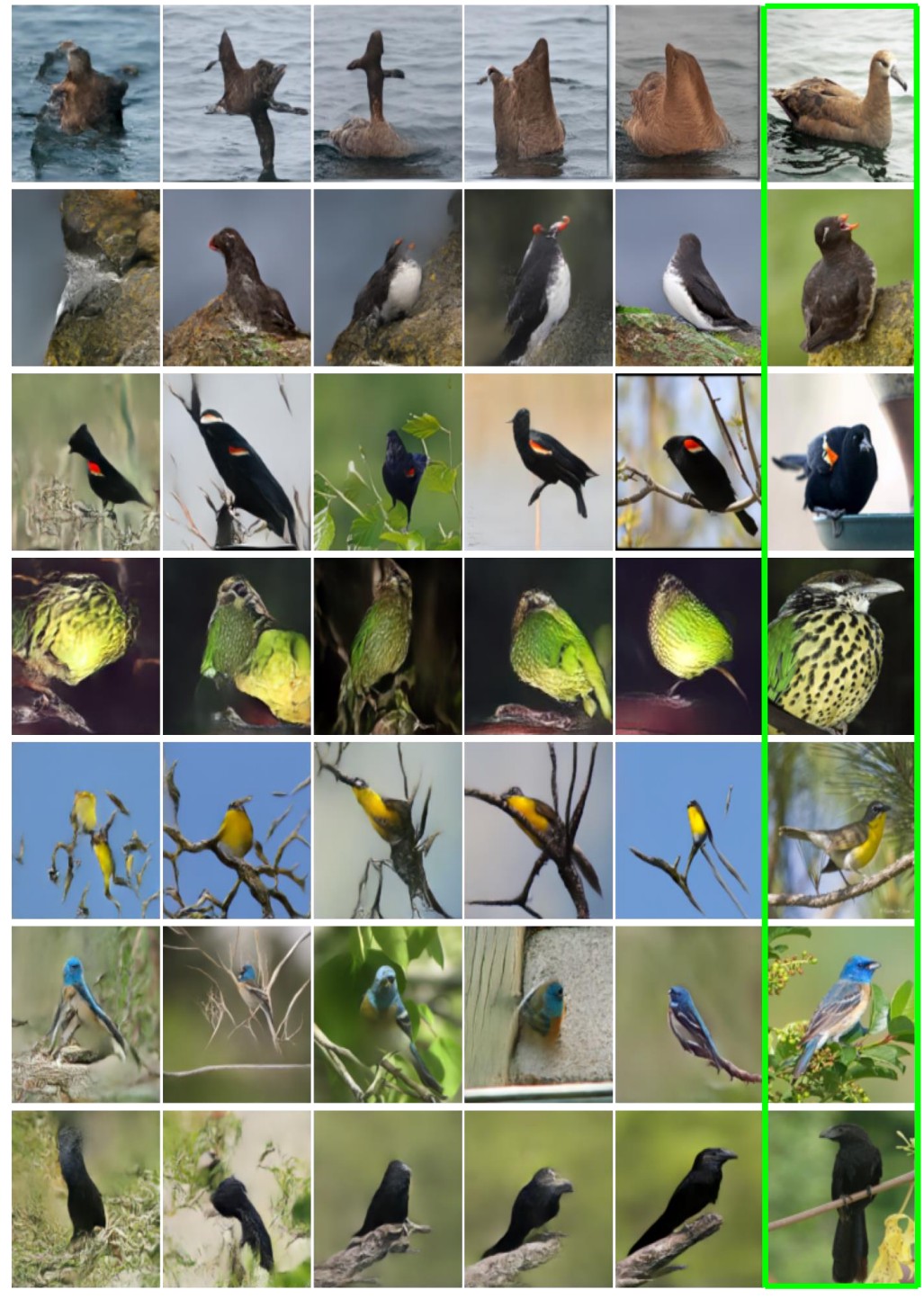

Figure 11: Generated anomaly images on the Caltech-UCSD Birds 200 dataset. The images highlighted in green are normal samples.

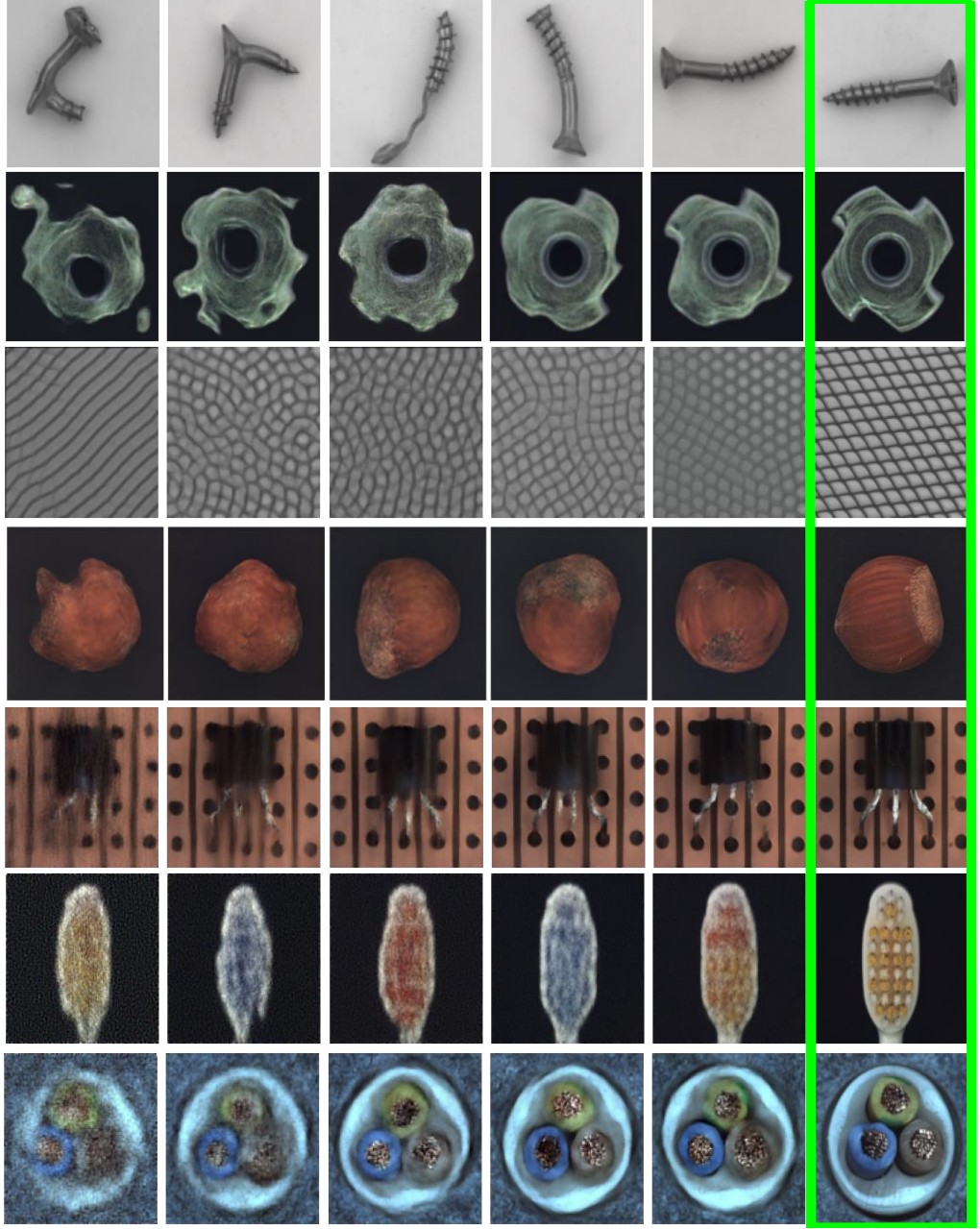

Figure 12: Generated anomaly images on the MVTecAD dataset. The images highlighted in green are normal samples.

## 11 DATASET DESCRIPTIONS

The setting is reported from (33).

**Standard Datasets**  We evaluate our method on a set of commonly used datasets: CIFAR-10 consists of RGB images of 10 object classes. CIFAR-100: we use the coarse-grained version that consists of 20 classes.

**Small datasets**  To further extend our results, we compared the methods on a number of small datasets from different domains: 102 Category Flowers and Caltech-UCSD Birds 200. For each of these datasets, we evaluated the methods using only each of the first 20 classes as normal, and using the entire test set for the evaluation. For FGVC-Aircraft, due to the extreme similarity of the classes to each other, we randomly selected a subset of ten classes from the entire dataset, such that no two classes have the same `Manufacturer`. Following are the selected classes: `[91,96,59,19,37,45 ,90,68,74,89]`.

MvTecAD: This dataset contains 15 different industrial products, with normal images of the proper products for training and 1 to 9 types of manufacturing errors as anomalies. The anomalies in MvTecAD are in-class, i.e. the anomalous images come from the same class of the normal images with subtle variations.

**Symmetric datasets**  We evaluated our method on datasets that contain symmetries, such as images that have no preferred angle (microscopy, aerial images): WBC : we used the 4 big classes in "Dataset 1" of the microscopy images of white blood cells, with a 80%/20% train-test split.

