# OpenReview forum: "Fake It Until You Make It : Towards Accurate Near-Distribution Novelty Detection"
_ICLR.cc/2023/Conference — ICLR 2023 poster_

### Official Review · Reviewer_mA9p · 2022-10-25

**Confidence:** 5
**Correctness:** 3
**Technical Novelty And Significance:** 2
**Empirical Novelty And Significance:** 3
**Recommendation:** 6

**Clarity, Quality, Novelty And Reproducibility:**

The paper is well written and easy to follow. However, there seem to be many components involved and many separate training steps and hyper-parameters that would make reproducibility hard.
The method is novel in general but all components are well explored before.

**Strength And Weaknesses:**

Strengths
1) Near anomalies are a very important case that should be tackled more.
2) Good insights and good results.
Weaknesses
1) I am not quite sure about the made distinction between Near ND and Near OOD. The authors stated the difference, however, without using any reference.
2) Design choices are not well explained, e.g., why a binary classifier is used, why NN is deployed for ND and not the binary classifier itself.
3) How about training complexity, hyper-parameter selection?

**Summary Of The Paper:**

The paper investigates the near anomalies detection and points a significant decrease in performance of state of the art approaches on near anomalies cases. A new solution is proposed: 1) using SDE models trained to generate data from the training distribution. 2) generate a dataset from the trained SDE. 3) train a binary classifier on top of a feature extractor to distinguish between the generated data and real data. 4) use nearest neighbour for detecting anomalies.
Extensive experiments and empirical analysis are carried on. The method shows good results.


**Summary Of The Review:**

A new method for detecting near anomalies based on SDE, binary classification training and then KNN.
Good empirical results.
Not super clear how design choices were made.

---

> ### Author Response · Authors · 2022-11-10
> **Response to Reviewr mA9p**
>
>
> Thank you for your review and useful comments. We have made a revision based on the reviewers' concerns, and specific comments are answered below.
>
> 1) The main distinction between Near Novelty Detection(ND) and Near Out-Of-Distribution(OOD) detection arises from the difference between ND and OOD. The common objective of OOD detection and ND is to detect anomalies. Since they share the same goal, they may be considered the same. However, in OOD detection an extra piece of information is present, which is the class information of each normal training sample. This extra information makes the techniques for the OOD detection and novelty detection quite different. For instance, one could achieve satisfactory detection rates in OOD, by simply training a model to classify normal classes, and threshold the maximum Softmax. This simple remedy is unfortunately impossible for novelty detection, where the training of the base classifier is impossible due to non-existence of this extra class information. Therefore, OOD detection methods cannot simply be used to solve the novelty detection problem. In light of this, the literature has categorized these two problems as being different [1,2]. We have clarified this difference between the OOD  and ND setting in the third paragraph of the introduction and we have added a reference (1,2) supporting that  these two problems are different (see paragraph 3 of page 2). To alleviate this concern, in the revised manuscript, we further added appendix 7.3,  in which this difference is discussed in more detail.
>
>
>
>
> 2)These are two reasons for this particular choice:
>   * Using nearest neighbors of pretrained networks features distance (NN) for novelty detection is a baseline that performs well on these tasks (e.g. PANDA, DN2, MSAD). The results would then be more comparable with them if such a choice was made in our method as well.
>   * The classifier has learned an auxiliary task of distinguishing normal versus specific fake generated images. This could be a little different from the general novelty detection. In fact, this auxiliary task is used to fine-tune the backbone, but using the nearest neighbors enables us to solve anomaly detection in a non-parametric and more flexible manner since pretrained models have a rich feature space. Furthermore, using the penultimate layer’s feature is better Than using the projection head for novelty detection. This is likely due to the penultimate layer preserving more information than the projection head, which has much smaller dimensions[3].
>
> 3)The most important hyper-parameter in our proposal is the stopping point of a generator model. We performed an ablation study and to address any concern related to stopping point, we have added appendix 7.4 to our revised manuscript. A list of other common hyper-parameters of our method (such as learning rate) can also be found in Appendix 7.
> Note that the training phase consists of three steps:
>   * step0: train the generator
>   * step1: generating synthesized samples
>   * step2: fine tuning the backbone
>
> As a result of the nature of diffusion-based models, step 1 has higher complexity. There are recent works aimed at speeding up the diffusion-based sampling, which could be explored in future works.
> Our model time complexity is also reported on a GTX 1080 on a variety of datasets:.
>
> |  | CIFAR10   | CIFAR100   | FGVC-Aircraft|Stanford-Cars  | WBC |   Weather| Birds  | Flowers |  MvTecAD|
> |:------:|:---------:|:---------:|:---------:|:----------:|:------:|:---------:|:---------:|:---------:|:----------:|
> |STEP0(Train the generator)|3h | 3h |3h|4h|2h | 5h |6h|6h|10h|
> |STEP1(Generating)|60h | 60h |20h|20h|15h | 20h |20h|20h|50h|
> |STEP2(Fine-Tuning)|5h | 5h |2h|2h|1h | 2h |2h|2h|4h|
>
> References:
>
> [1]  Ruff L, Kauffmann JR, Vandermeulen RA, Montavon G, Samek W, Kloft M, Dietterich TG, Müller KR. A unifying review of deep and shallow anomaly detection. Proceedings of the IEEE. 2021 Feb 4;109(5):756-95
>
> [2] Salehi, Mohammad Reza, et al. "A unified survey on anomaly, novelty, open-set, and out-of-distribution detection: Solutions and future challenges." Transactions on Machine Learning Research(TMLR) (2022).
>
> [3] Sun, Yiyou, et al. "Out-of-distribution Detection with Deep Nearest Neighbors." International Conference on Machine Learning (ICML) 2022

---

> ### Author Response · Authors · 2022-11-28
> **A friendly reminder of the rebuttal conclusion**
>
> Your valuable comments are greatly appreciated. Due to the closing date of the open discussion, we would like to kindely remind you to read our response, which hopefully will address all your concerns. We would love to hear your feedback and also love to answer if you have additional questions.

---

### Official Review · Reviewer_wBQG · 2022-10-25

**Confidence:** 4
**Clarity, Quality, Novelty And Reproducibility:** Please see the Strength and Weaknesse…
**Correctness:** 4
**Technical Novelty And Significance:** 3
**Empirical Novelty And Significance:** 3
**Recommendation:** 6

**Strength And Weaknesses:**

* There is a large room for improvement in clarity.
    * The paper should be clearer on how exactly the generative model is trained and used. The most important fact, in my opinion, is that the score-based generative model is trained on in-distribution data and terminated prematurely. However, the fact is only briefly mentioned in Page 2, and then mentioned in Figure 2. Given the importance of this information, this point should be described in detail in Section 2.1. Also, other details and design choices, for example, how early the training of the score-based model should be stopped, needs to be elaborated.
    * The experimental setup described in Section 3.2 is difficult to follow. There are several comparison settings but are not well organized.
* Discussion on the differences among diffusion models, GANs, and VAEs in Section 2.2 are not very persuasive. It does not have concrete examples or quantitative arguments. Plus, the argument of Section 3.2 is mostly about generative modeling itself and not about its application to generating auxiliary data in novelty detection.
* When performing CIFAR-10 vs CIFAR-100 experiment, it is more standard and more challenging to incorporate all the classes, instead of using only one class from each dataset. Overall, I am not convinced that the experiment settings used in the paper are standard.
* Replacing a GAN with a diffusion model in the data generator for novelty detection itself is not a very novel contribution to the community.

**Summary Of The Paper:**

The paper proposes a novelty detection method that utilizes synthesize outliers during training. The synthetic outliers are generated from a score-based generative model trained on in-distribution data.

**Summary Of The Review:**

Even though the problem that the paper tackles is relevant to the community, the paper has limited novelty and clarity.

---

> ### Author Response · Authors · 2022-11-10
> **Response to Reviewr wBQG**
>
> Thank you for your review and useful comments. We have made a revision based on the reviewers' concerns, and specific comments are answered below.
>
> 1_1)The fake data is generated using an early-stop generator. There are numerous places where we discuss our proposed strategy (Last paragraph of Step 1 in section 2.1, Figure 3, section 2.2 ). We understand the concern regarding the stopping point of the generator and other details, The stopping point effect was examined in the ablation study (table-4). Further details and design choices are provided in section 3 and appendix 7. To alleviate this concern, in the revised manuscript, we added appendix 7.4 which explores this stopping point in more detail.
>
> 1_2)We have made a revision based on the reviewers' concern. We kindly ask the reviewer to retake a look at this Section 3 In which Our main results were organized into three main categories:
>
>   * a) A study of the results of the method on regular datasets is in the anomaly detection literature. These results indicate how our method  performs on the common novelty detection setup. See table 1(a).
>
>   * b) The results of the method on datasets with classes that are semantically close to each other but are not common in novelty detection works. These results indicate our performance on the near novelty detection setup. See Table 1(b).
>
>    * c)The results of the method on settings that were not explored previously, such as CIFAR10-vs100 in the class-wise setting. These results are also indicative of improved performance on the near novelty detection setup. See Table 1(c).
>
> 2)Incorporating all classes in the CIFAR-10 vs CIFAR-100 experiment is common in the setting of Out-of-Distribution Detection [1], which assumes labels are available during training. Instead, we consider the Novelty Detection setting, where we only have access to one class of training data during training [2]. We discussed the difference between the Out-of-Distribution and Novelty Detection setting in the third paragraph of the introduction, and we will clarify it further.
>
> References:
>
> [1]Salehi, Mohammadreza, et al. "A unified survey on anomaly, novelty, open-set, and out-of-distribution detection"
>
> [2] Ruff L, et al A unifying review of deep and shallow anomaly detection.
>
> 3)In section 2.2,  we provided the rationale on why early stopping, which is a critical element of our proposed fake anomaly generation, only works in certain generative models (such as the diffusion model), and fails otherwise (e.g. GANs). This argument is consistent with the empirical evidence that the early-stopped diffusion model produces near-distribution samples, which fits our needs.
> We gave two concrete examples here. One was provided in Fig. 3 left column, where GANs show a pretty noisy and non-monotonic trend of FID. The other is more quantitative and consists of testing out how other models aimed at generating auxiliary data in novelty detection works in our setup. Here, we have considered OpenGAN, which is a SOTA model in the out-of-distribution settings, and adjusted it for our setting (i.e. novelty detection).  (see Table 1-c). In addition, we conducted an ablation study in Table 4, which evaluates many SOTA generative models against the SDE-based models based on AUROC, density, and coverage metrics.
>
> 4)This study looked at novelty detection from a new perspective. The Near-ND setup has not existed before, and emerged in our paper. We also did extensive experimental&theoretical validation of our methods that show a significant improvement in this setup compared to SOTA ND. Indeed, we raised a significant failure mode of existing novelty detection schemes, which comes up in many real world applications (e.g. face liveness detection), and provided an efficient solution for it: (i)Defining a new novelty detection setup “Near-ND”: We empirically showed that SOTA novelty detection methods show a significant detection performance drop when the anomalies have similarities with the normal data (see table 1-c)  (ii)Novel method for generation of training anomaly samples: We proposed to use SDE as a powerful tool for generation of anomaly training data to boost existing algorithms. The key challenge here is that we have only access to the normal data in training of the generative model. The fact that the generated data needs to be both diverse and high quality makes the problem more challenging. We showed that early-stopping of SDEs could be used as an effective strategy for this purpose. There we showed that this strategy is specific to SDEs, and GANs do not necessarily exhibit this property when their training is early-stopped. We also tested this proposition empirically. (iii)Training strategy and evaluations: To address the shortcomings of previous methods in the near-ND setup, we have proposed to use the generated anomalous data during training. We did extensive evaluations and ablation studies to show the generality of the proposed method.

---

> > ### Comment · Reviewer_wBQG · 2022-11-28
> > **Thank you for your response**
> >
> > Dear authors,
> >
> > First of all, thank you for your effort in elaborating and updating the manuscript.
> >
> > As all the other reviewers mentioned, the arbitrariness of the early stopping moment is the biggest weakness of the proposed algorithm. I doubt that FID range can be a reliable guideline because FID tends to have different scales for different types of images. The paper should be stronger if it can provide a deeper understanding of exactly what aspect of the generated images improves novelty detection and come up with a better metric to determine the stopping point.
> >
> > My biggest remaining concern is the use of single-class normal data in all of the experiments. Testing the algorithm under the very limited variability of normal data makes the scalability of the proposed method with respect to the diversity of normal data questionable. The diversity in the normal data is another factor that makes the novelty detection problem difficult, and it is important to explore how the proposed algorithm behaves under this condition. For example, CSI provided the experimental results on multi-class normal data in their paper.
> >
> > Since addressing these two points requires a major update of the manuscript, and thus the current version is not quite qualified for presentation at the conference.

---

> > > ### Author Response · Authors · 2022-11-29
> > > **FID After Convergence for SDE**
> > >
> > > The following is a detailed report of the training results of our generative backbone (SDE) after convergence in order to clarify the FID scale for different datasets and images. Using a multiclass dataset of **C** classes, we train SDE separately for each class, and evaluate the models after convergence, as shown in table[1-9]. For each dataset, the average of the **C** classes was also reported, and table-10 provided an overview of all 9 datasets.
> > >
> > > | CIFAR-10  |   |   | |  |  |   |   |  |  |  |     |
> > > |:------:|:----:|:---:|:------:|:----:|:------:|:---:|:----:|:-----:|:----:|:------:|:------:|
> > > |Class Number|0 | 1 |2|3|4 | 5 |6|7|8|9 | average  |
> > > |FID (After Convergence) |  6.3 | 5.7 | 9.7|10.5| 9.5 | 7.2 |10.0|8.0|8.2|7.8 | **8.29** |
> > >  (Table 1)
> > >
> > >
> > > | CIFAR-100  |   |   | |  |  |   |   |  |  |  |      |   |   |   | |  |  |   |   |  |  |
> > > |:------:|:---------:|:---------:|:---------:|:----------:|:------:|:---------:|:---------:|:---------:|:----------:|:------:|:------:| :------:|:---------:|:---------:|:---------:|:------:|:------:|:---------:|:---------:|:---------:|:----------:|
> > > |Class Number|0 | 1 |2|3|4 | 5 |6|7|8|9 | 10  |11|12 | 13 |14|15|16 | 17 |18|19|average|
> > > |FID ( After Convergence) |  10.2|8.3|9.9|9.4|9.8|7.5|9.5|8.5|7.9|7.6|8.0|6.8|10.2| 9.8|7.3|8.9|9.0|9.4|8.9|8.3|**8.76**
> > >  (Table 2)
> > >
> > >
> > > |Flowers |   |   | |  |  |   |   |  |  |  |      |   |   |   | |  |  |   |   |  |  |
> > > |:------:|:-------:|:-------:|:--------:|:-----:|:-----:|:-------:|:---------:|:---------:|:----------:|:------:|:------:| :------:|:---------:|:---------:|:---------:|:----------:|:------:|:---------:|:---------:|:---------:|:----------:|
> > > |Class Number|0 | 1 |2|3|4 | 5 |6|7|8|9 | 10  |11|12 | 13 |14|15|16 | 17 |18|19|average|
> > > |FID ( After Convergence) | 37.0 | 40.6 | 35.2 | 43.5 | 39.1 | 50.8 | 44.6 | 39.4 | 41.6 | 46.0 | 42.1 | 35.7 | 42.4 | 42.1 | 39.7 | 48.7 | 48.9 | 37.6 | 46.1 | 45.8 |**42.35**
> > >    (Table 3)
> > >
> > >
> > >
> > >
> > > |Birds |   |   | |  |  |   |   |  |  |  |      |   |   |   | |  |  |   |   |  |  |
> > > |:------:|:------:|:---------:|:----:|:-------:|:------:|:------:|:---------:|:---------:|:----------:|:------:|:------:| :------:|:----:|:----:|:----:|:----------:|:------:|:---------:|:----:|:---------:|:----------:|
> > > |Class Number|0 | 1 |2|3|4 | 5 |6|7|8|9 | 10  |11|12 | 13 |14|15|16 | 17 |18|19|average|
> > > |FID ( After Convergence) | 38.5 | 35.1 | 27.4 | 33.6 | 39.8 | 32.6 | 39.3 | 45.3 | 30.7 | 52.1 | 52.0 | 45.9 | 47.8 | 48.2 | 36.1 | 26.4 | 48.5 | 47.0 | 35.2 | 26.3 |**39.40**
> > >   (Table 4)
> > >
> > >
> > > |Stanford-Cars |   |   | |  |  |   |   |  |  |  |      |   |   |   | |  |  |   |   |  |  |
> > > |:------:|:---------:|:---------:|:---------:|:----------:|:------:|:---------:|:---------:|:---------:|:----------:|:------:|:------:| :------:|:---------:|:---------:|:---------:|:----------:|:------:|:---------:|:---------:|:---------:|:----------:|
> > > |Class Number|0 | 1 |2|3|4 | 5 |6|7|8|9 | 10  |11|12 | 13 |14|15|16 | 17 |18|19|average|
> > > |FID ( After Convergence) |  51.9 | 55.7 | 35.8 | 56.8 | 37.9 | 32.4 | 36.6 | 54.8 | 33.5 | 49.5 | 61.3 | 33.2 | 55.9 | 46.9 | 59.5 | 55.0 | 49.1 | 37.7 | 62.1 | 39.6 |**47.26**
> > >   (Table 5)
> > >
> > > |MvTecAD |   |   | |  |  |   |   |  |  |  |      |   |   |   | |   |  |
> > > |:------:|:---:|:----:|:---:|:---:|:-----:|:-----:|:---------:|:---------:|:----------:|:------:|:------:| :------:|:---------:|:---------:|:---------:|:----:|:------:|
> > > |Class Name|Bottle| Hazelnut| Capsule |Metal| Nut| Leather| Pill| Wood| Carpet| Tile| Grid |Cable |Transistor| Toothbrush |Screw| Zipper| average|
> > > |FID ( After Convergence) |  31.5 | 34.7 | 42.7 | 40.1 | 37.9 | 30.5 | 35.4 | 33.6 | 44.3 | 41.9 | 43.4 | 31.7 | 34.8 | 40.0 | 44.6 | 33.8 |**35.48**
> > >   (Table 6)
> > >
> > >
> > > | FGVC-Aircraft |   |   | |  |  |   |   |  |  |  |     |
> > > |:----:|:------:|:---:|:----:|:-----:|:----:|:------:|:----:|:-----:|:-----:|:------:|:------:|
> > > |Class Number|0 | 1 |2|3|4 | 5 |6|7|8|9 | average  |
> > > |FID ( After Convergence) |  33.2 | 37.7 | 34.4 | 44.1 | 32.8 | 34.1 | 40.8 | 34.5 | 37.3 | 43.9 | **37.28** |
> > >   (Table 7)
> > >
> > >
> > >
> > >
> > > |   WBC|   |   |   |  | |
> > > |:------:|:---------:|:---------:|:-----:|:------:|:-----:|
> > > |FID| 1 | 2 | 3 | 4 |average|
> > >  |FID ( After Convergence) | 43.6 | 41.9 | 48.2 | 50.6 | **46.07**|
> > > (Table 8)
> > >
> > >  |   Weather|   |   |   |  | |
> > > |:------:|:----:|:----:|:---:|:----:|:----:|
> > > |Class Name| shine|sunrise|cloud|rain|average|
> > >  |FID ( After Convergence) | 40.1 | 36.4 | 44.7 | 34.3 | **38.87**|
> > > (Table 9)
> > >
> > > * * *
> > >
> > >
> > >  | Resolution |   32x32|    32x32| 128x128| 128x128 | 128x128 |  128x128 | 128x128  | 128x128 | 128x128 |
> > > |:--:|:---:|:----:|:---:|:----:|:---:|:----:|:---:|:-------:|:--:|
> > > |Datasets|CIFAR-10 | CIFAR-100 |Birds|Flowers|MvTecAD | Weather|WBC|FGVC-Aircraft|Stanford-Cars|
> > > |FID ( After Convergence)|  **8.29** | **8.76** | **39.40** | **42.35** |**35.48** |  **38.87** |**46.07** | **37.28** | **47.26** |
> > > | Stopping Point$\approx$FID-After-Convergence $\times5$ | 40|    40| 200| 200|200| 200 | 200  |200 | 200 |
> > >  (Table 10)

---

> > > > ### Comment · Reviewer_wBQG · 2022-11-29
> > > > **Response**
> > > >
> > > > Dear authors,
> > > >
> > > > Thank you so much for the quick update. I admire your massive effort on this.
> > > >
> > > > Your additional results resolve my concerns and I would update my ratings accordingly. It is very interesting to watch that the proposed method works well under the multi-class unlabelled setting which looks promising. I strongly encourage authors to emphasize this setting in the camera-ready version of the paper. Even though the novelty detection methods have been focusing on the single-class setting, we need to move forward.

---

> > > > > ### Author Response · Authors · 2022-11-29
> > > > > **Response**
> > > > >
> > > > > Thank you, we're glad to hear your concerns were resolved! We appreciate you bringing this setting to our attention, We indeed plan to include multi-class unlabelled setting to the paper.

---

> > > ### Author Response · Authors · 2022-11-29
> > > **Response to Reviewr wBQG**
> > >
> > > Thank you so much for taking the time to review and comment. specific comments are answered below:
> > >
> > > > **Comment: I doubt that FID range can be a reliable guideline because FID tends to have different scales for different types of images.**
> > >
> > > **Response:**
> > > Our experiments across nine different types of datasets (for emaple medical images, object classification, and quality control,etc.) and their different classes showed that datasets of the same resolution had roughly the same FID metric after SDE convergence.
> > >
> > >
> > > > **The paper should be stronger if it can provide a deeper understanding of exactly what aspect of the generated images improves novelty detection and come up with a better metric to determine the stopping point.**
> > >
> > > **Response:**
> > > Our experiments and theoretical studies demonstrate that the generated images by SDE are suitable for Novelty detection and we demonstrate its superiority through metrics such as FID, diversity, and coverage. The diversity and noiseless characteristics of generated images are important, as we discussed. Furthermore, As a result of our experiments for novelty detection and near-novelty detection across   different settings and different classes of them, we found that the FID metric has roughly the same scale depending on the resolution of training samples.
> > >
> > > > **Comment: I doubt that FID range can be a reliable guideline because FID tends to have different scales for different types of images.**
> > >
> > > **Response:**
> > > Based on the FID of SDE after convergence, which can be found with details in table[1-10], we fixed the stopping point. As a rule of thumb, we set stopping point to 5 times the converaging FID, which results in FID $\approx$ 40 and 200, respectively, for low- and high-resolution.
> > > In addition, based on our ablation study in appendix 7.3 and table 4, the stopping point is not fragile and is a reliable parameter. Table 11-12 contains the details of the result of ablation study.
> > > we would appreciate it if you would take a look at our additional appendix and our response to other reviewers about stopping point.
> > >
> > > | CIFAR-10vs100(Low-resolution) |   |   | |  |  |   |   |  |  |  |   |   |   |  |
> > > |:------:|:---------:|:---------:|:---------:|:----------:|:------:|:---------:|:---------:|:---------:|:----------:|:------:|:---------:|:---------:|:---------:|:----------:|
> > > |FID|400 | 350 |300|250|200 | 150 |100|60|50|40 |30 |20|15|10|50|40|
> > > |AUC|82.8 | 83.1 |83.4|83.6|86.0 | 87.3 |88.5|88.6|89.2|90.0 |90.0 |86.9|78.5|68.2|
> > > (Table 11)
> > >
> > >
> > > | WBC(High-resolution)  |   |   | |  |  |   |   |  |  |  |   |   |   |  | | |
> > > |:------:|:---------:|:---------:|:---------:|:----------:|:------:|:---------:|:---------:|:---------:|:----------:|:------:|:---------:|:---------:|:---------:|:----------:|:----------:|:----------:|
> > > |FID|500 | 350 |300|270|250 | 240 |220|200|180|160 |140 |120|100|75|50|40|
> > > |AUC|76.2 |80.7 |84.6|88.5|89.5 |90.4 |91.0|91.2|91.1|91.2 |89.6 |90.5|86.3|77.9|74.3|70.6|
> > > (Table 12)
> > >
> > >
> > > > **My biggest remaining concern is the use of single-class normal data in all of the experiments. Testing the algorithm under the very limited variability of normal data makes the scalability...**
> > >
> > > **Response:**
> > >
> > > Based on the longer line of literature describing novelty detection, we focused on the one-class setting, which uses one class of a dataset as a normal sample source. However, we believe that unlabeled-multi-class could be addressed with minor revisions in our manuscript, and we can explore that also in our appendix.
> > > In response to your concern, we provide three experiments and compare our method with CSI in table 11. This indicates our validity in this setting as well. The details of the experiment are given below.  Our experiment will be extended in the near future to include additional datasets
> > >
> > > **Unlabeled multi-class**: In this setup, we assume that normal samples are from a specific multi-class dataset (i.e. CIFAR-10) without labels, testing on various external datasets as abnormal samples(e.g. CIFAR-100). We copied the CSI results from the original paper.
> > >
> > > |    |   | AUROC (%) of methods trained on unlabeled CIFAR-10 |   |
> > > |:------:|:---------:|:---------:|:---------:|
> > > |    |  SVHN |LSUN| CIFAR-100  |
> > > |Ours| **99.9** | **99.4** | **95.7** |
> > >  |CSI | 99.8 | 97.5 | 89.2 |
> > >   (Table 13)

---

> ### Author Response · Authors · 2022-11-28
> **A friendly reminder of the rebuttal conclusion**
>
> We sincerely appreciate your detailed feedback on our work. We have responded to each of your concerns. Please let us know if your concerns have been addressed. If you have any further questions, we are more than happy to address them before the conclusion of the rebuttal phase.

---

### Official Review · Reviewer_xzvz · 2022-10-31

**Confidence:** 4
**Correctness:** 3
**Technical Novelty And Significance:** 2
**Empirical Novelty And Significance:** 3
**Recommendation:** 6

**Clarity, Quality, Novelty And Reproducibility:**

This paper is very clearly written and easy to understand. I really enjoy reading this paper  and well-designed figures help me a lot to  understand its motivation and the proposed framework.




**Strength And Weaknesses:**

The idea of this paper is very natural: try to generate near-distribution outliers to augment the original training dataset which only contains normal data (semi-supervised learning), such that we can fine-tune the pre-trained feature extractor by training a binary classifier.

Strengths:
(1) Using generated outliers to improve the accuracy of novelty detection is not new. There have been some similar works which use GANs instead of SDE.  The main contribution of this paper is that they replace the GANs with the diffusion model (which is increasingly popular recently).  The motivation here is that the outliers generated by SDE are “closer“ to the nominal distribution, and then using these "boarder" points  to deal with the near-ND setting.  Even though the pipeline and SDE are all from existing works, the novel combination is meaningful considering the theoretically and experimental results.

(2) The experimental results show that the proposed method considerably improves over existing models (especially for the near-ND setting).  It is not surprising that SDE over-perform GANs for the near-ND setting, but I am surprised that the improvement is so obvious.  It will be great if the code can be provided.

Weakness:
(1) In Section 2.2 (Diffusion models vs. GANs), they try to explain, for near-ND, why choosing diffusion models instead of other generative modes (GAN and VAE), it would be better to include flow-based models in this part. Also,  VAE-related results are not included in figure3.

(2) In Figure3,  density, coverage and FID are used as the metrics to measure the fidelity and diversity.  Could you add a brief introductions/definitions for these metrics? Also, I am confused how to get the "density" from GAN-based models as they are not tractable models.

(3)About Stopping point: In section 4, "the method is robust against the stopping point with at most 2% variation for the reasonable FID scores". However,  from table4,  it is difficult to get this conclusion. what is "reasonable" FID? Actually, the stopping point decide how similar the fake points and the real points. How to decide when to stop ?

Other minor comments:
The "normal distribution" used in this paper sometimes are confusing, which means the distribution of nominal data instead of Gaussian distribution. It would be better to use other notations.


**Summary Of The Paper:**

This paper is focused on a challenging task: near-distribution novelty detection (near-ND). The authors propose to exploit a diffusion generative model to produce synthetic near-distribution anomalous data(i.e., “fake"), then the  pre-trained feature extractor is  fine-tuned to distinguish generated anomalies from the normal samples.


**Summary Of The Review:**

Overall, this paper sheds some insights that diffusion- based data generation considerably passes the GAN-based SOTAs in the novelty detection task, especially for the neat-ND setting.  I recommend to accept this paper.

---

> ### Author Response · Authors · 2022-11-10
> **Response to Reviewr xzvz**
>
> Thank you for your review and useful comments. We have made a revision based on the reviewers' concerns, and specific comments are answered below.
>
> 1)Our new submission includes the VAE-based results/samples in figure 3. It is common for flow-based methods to perform poorly compared to GANs and diffusion-based models in image generation. For instance, a SOTA flow-based generative model [1] achieved the FID of 34.4 on CIFAR-10, where the SOTA GANs FID on CIFAR-10 is 1.8 [2]. Because of this, we did not include flow-based methods in our theoretical/empirical results. A new experiment was conducted to address the reviewed comment-1 and the results were included in table 3 of the revised manuscript. As shown below, the results are:
>
> |   |   | Datasets |   |  |
> |:------:|:---------:|:---------:|:---------:|:----------:|
> | | CIFAR10 | Weather | WBC | MvTecAD |
> |DenseFlow[1]| 96.4 | 84.9|82.0 | 78.3  |
>  |SDE | 99.1 | 97.0 | 91.2 | 86.4 |
>
> References:
>
> [1] Grcić M, Grubišić I, Šegvić S. Densely connected normalizing flows. Advances in Neural Information Processing Systems.
> [2] Sauer, Axel, Katja Schwarz, and Andreas Geiger. "Stylegan-xl: Scaling stylegan to large diverse datasets."
>
> 2)This metric has been clarified in our new submission and a section has been added to the Appendix to provide more details on Density&Coverage and its equation(see Appendix 7.1). The density and coverage metrics can be used to assess the diversity and fidelity of generative models, respectively. The method measures the distance between the generated and real images using a manifold estimation procedure.In order to exploit feature representations for these metrics, features prior to the final classification layer of a pre-trained model(e.g. Inception V3) are used. Note that the estimation relies solely on generated and real data, so they could be applied to any sampling based generative model.
>
> 3)As a result of visual inspection of training images, it is evident that FID ranges of 30-50 and 150-250, are semantically different from the normal class for low- and high-resolution images, respectively. It is important to note that this range appears to be universal, as it is consistent across a wide range of datasets.  Clearly, one should not expect reasonable results for the entire range of FIDs. With very low FID values, the generated images are indistinguishable from the normal class, which is not intended. On the other hand, with very high FID values, the generated image would be very noisy and do not also necessarily capture the near-distribution concept.  For the latter case, one has to note that current deep models are highly capable of learning artifacts from their input, instead of paying attention to the general semantics of the input. Therefore, high FID samples would be useless in making the network learn non-trivial features in detecting anomalies.  However, we noted that the range of FIDs that work is quite wide and universal, so one could design a rule of thumb for it. We have added appendix 7.3 to our revised manuscript, which explores the stopping point in more detail.
>
> | CIFAR-10vs100(Low-resolution) |   |   | |  |  |   |   |  |  |  |   |   |   |  |
> |:------:|:---------:|:---------:|:---------:|:----------:|:------:|:---------:|:---------:|:---------:|:----------:|:------:|:---------:|:---------:|:---------:|:----------:|
> |FID|400 | 350 |300|250|200 | 150 |100|60|50|40 |30 |20|15|10|50|40|
> |AUC|82.8 | 83.1 |83.4|83.6|86.0 | 87.3 |88.5|88.6|89.2|90.0 |90.0 |86.9|78.5|68.2|
>
> | WBC(High-resolution)  |   |   | |  |  |   |   |  |  |  |   |   |   |  | | |
> |:------:|:---------:|:---------:|:---------:|:----------:|:------:|:---------:|:---------:|:---------:|:----------:|:------:|:---------:|:---------:|:---------:|:----------:|:----------:|:----------:|
> |FID|500 | 350 |300|270|250 | 240 |220|200|180|160 |140 |120|100|75|50|40|
> |AUC|76.2 |80.7 |84.6|88.5|89.5 |90.4 |91.0|91.2|91.1|91.2 |89.6 |90.5|86.3|77.9|74.3|70.6|

---

> ### Author Response · Authors · 2022-11-28
> **A friendly reminder of the rebuttal conclusion**
>
> Our sincere thanks go out to you for your precious comments. Since we are closing to the end date of the open discussion, we would like to kindly remind you to read our response which hopefully should resolve all your concerns. We would love to hear your feedback and also love to answer if you have additional questions.

---

### Official Review · Reviewer_ahZi · 2022-10-31

**Confidence:** 3
**Correctness:** 3
**Technical Novelty And Significance:** 3
**Empirical Novelty And Significance:** 3
**Recommendation:** 6

**Clarity, Quality, Novelty And Reproducibility:**

- Despite the simplicity of the proposed method, the proposal to use
early-stopped diffusion models to generate the OOD data is novel to my knowledge; this proposal is
justified soundly both empirically and theoretically (albeit with intuition rather than rigorous
proof).
- While there are some typographic errors and some slightly awkward phrasing, the paper is generally
strong in terms of clarity -- the explanation and rationale of the method (supported greatly by the
figures) is easy to understand as is the evaluation procedure.
- In terms of reproducibility, the paper and its appendices provide detailed descriptions of the
training/evaluation setup. Experimental (optimisation + model) settings are provided in the
appendices and code is provided as part of the supplementary material.


**Strength And Weaknesses:**

### Strengths

- The paper is, for the most part, well-written, easy-to-follow, and does a good job of
contextualising and motivating the work.
- The proposed method is simple and intuitive, and seems to perform well empirically
compared with the baselines in both the regular ND setting and the near ND setting, showing a
significant reduction in the performance gap of going from the former to the latter.
the NR to near-NR setting.
- The figures are well-chosen and well-put-together: they convey well the essential points of the
  paper, such as the shortcomings of existing methods and the feasibility of using GANs vs diffusion
  models for creating the synthetic data.
- Evaluation performed using a good range and selection of baseline methods and benchmark datasets.
- Results are aggregated over a good number (10) of replicates.

### Weaknesses
- Despite the authors claiming otherwise, the method is quite sensitive to the stopping criterion in
  the near ND setting with an empirical range of [68.2, 90.0]; in practice, different datasets may
  require wildly different FID thresholds and there is no reliable way of validating this given the
  nature of the problem. Also, I wonder if the number of sampling steps and the choice of sampler have a
  significant effect on the results -- there doesn't seem to be any mention of this.
  Taking this a step further, might it be feasible to train the diffusion model fully and then rely
  upon early-stopping the reverse diffusion process according to the FID score?
- The need to train, and indeed sample, from a diffusion model can be computationally burdensome for
  higher-resolution datasets.
- The description of the SDE model is a little overdone; a much briefer description with a pointing of the
  reader to the original paper would be sufficient. This is especially the case given that, as
  I understand it, the proposed method should work with any diffusion based model (Imagen, Parti,
  DALLE-2, etc.) or are there theoretically some constraints on the form said model and the
  associated sampler should take?
- While this applies to several methods of the same class, the method seems to make certain
assumptions about the distribution of the ID and OOD data that might not hold in the wild.
- While results are aggregated over a good number (10) of replicates no measure of dispersion
  (standard deviation/error) seems to be reported alongside the mean in each case.
- Some typographic and syntactical errors (e.g. 'that unlikely come' (line 1 of the introduction)).


**Summary Of The Paper:**

The paper tackles the problem of __near__ novelty detection (near ND), with __near__ referring to
the case in which the novelty (OOD) classes derive are semantically similar to those classes
contained in the in-distribution (ID) set used for training (a novelty class under this regime, for
example, being 'fox' when the ID set contains 'dog'). The authors begin by showing that
recently-proposed ND algorithms exhibit a significant performance gap with respect to the standard (far)
ND vs. near ND setup to motivate the problem. To solve this, the authors propose to use
early-stopped diffusion models, trained on the ID data, to generate a dataset of anomalous samples, with
features for the ND scoring mechanism can be learned by framing the task as one of binary
discrimination (real data vs. diffusion-model-generated data), solved with a linear classifier
head. In order to score samples, a simple k-NN-based algorithm is employed, acting on the aforementioned
features extracted from the training set. The authors show that this simple algorithm succeeds in reducing the gap between near ND and ND performances while also improving upon previous methods in the far ND setting. It is reasoned and shown empirically GANs (as in OpenGAN) lack some the necessary properties that diffusion models that are crucial to this success (e.g. image-quality metrics do not show a monotonic improvement in generative ability as training progresses).


**Summary Of The Review:**

The paper is well-motivated and proposes a simple but apparently strong method for tackling the
problem of near novelty-detection, a setting in which the novel (OOD) classes are semantically
close to classes in the training set. The explanation of the method is easy-to-follow and the
problem and the solution are soundly established. The results look promising and have been compiled across a good range of datasets and baselines. While I worry that the sensitivity of the
method to the stopping criterion, as well as the assumptions
that are made in the distributions of the ID/OOD datasets, might limit practical applicability, I nonetheless think the
overall the paper is strong enough to meet the threshold for acceptance based on the aforementioned merits.

---

> ### Author Response · Authors · 2022-11-10
> **Response to Reviewr ahzi**
>
>
> Thank you for your review and useful comments. We have made a revision based on the reviewers' concerns, and specific comments are answered below.
>  1)
> | CIFAR-10vs100(Low-resolution) |   |   | |  |  |   |   |  |  |  |   |   |   |  |
> |:------:|:---------:|:---------:|:---------:|:----------:|:------:|:---------:|:---------:|:---------:|:----------:|:------:|:---------:|:---------:|:---------:|:----------:|
> |FID|400 | 350 |300|250|200 | 150 |100|60|50|40 |30 |20|15|10|50|40|
> |AUC|82.8 | 83.1 |83.4|83.6|86.0 | 87.3 |88.5|88.6|89.2|90.0 |90.0 |86.9|78.5|68.2|
>
> | WBC(High-resolution)  |   |   | |  |  |   |   |  |  |  |   |   |   |  | | |
> |:------:|:---------:|:---------:|:---------:|:----------:|:------:|:---------:|:---------:|:---------:|:----------:|:------:|:---------:|:---------:|:---------:|:----------:|:----------:|:----------:|
> |FID|500 | 350 |300|270|250 | 240 |220|200|180|160 |140 |120|100|75|50|40|
> |AUC|76.2 |80.7 |84.6|88.5|89.5 |90.4 |91.0|91.2|91.1|91.2 |89.6 |90.5|86.3|77.9|74.3|70.6|
>   * As the results demonstrate, choosing a stopping point in the beginning and end stages of training of a generator leads to poor results. It is important to note that the synthesized images were not an effective anomaly in the first training procedure because they are extremely noisy. Furthermore, one has to note that current deep models are highly capable of learning artifacts from their input, instead of paying attention to the general semantics of the input. In addition, once the model converges to the real distribution, the images generated become similar to the normal distribution, which disallows using these images as anomaly samples. Therefore, it seems reasonable that performance results in the first and last phases of the training procedure would be poor.  indicating that the method is relatively robust against the limited variations in stopping points and the results are not fragile. Here are some results for stopping point 40 and 200 FID (low- and high-resolution). According to Table 4 and this more detailed result, the method is relatively robust against limited variations in stopping points. In addition, the range of FIDs that work seems to be quite wide and universal across datasets. Our results indicate that a FID range of 30-50 and 150-250 works across all datasets with low- and high-resolution images, respectively. We have added appendix 7.3 to our revised manuscript, which explores the stopping point in more detail.
>
>
>
>   * Our new submission clarifies the hyper-parameters regarding the sampling steps and sampler choice in appendix 7. Hyper-parameters were determined using the source code default (for example, predictor-corrector as a sampler). Considering that Score-based generative models use a denoising process to generate samples, stopping the reverse diffusion process would result in generating noisy samples, which, as discussed in the previous part, can lead to poor pretraining of the backbone.  Also our empirical verification on some datasets showed that other samplers (e.g. ODE-sampler) yield the same results and their generated images are perceptually indistinguishable with the one that was originally used in our experiments. Our new submission further clarifies the hyperparameters regarding the sampling steps and sampler choice in appendix 7.
>
> 2)Since we stop the training of the generator before convergence, the training would be less complex, and would take less time than a completely trained generator.The training phase consists of three steps:
> step0: generator training
> step1: generating synthesized samples
> step2: fine tuning the backbone
> Based on a GTX 1080 GPU, the average training time for each datasets was 4 hours for Step0, 30 hours for Step1, and 3 hours for Step2. As a result of the nature of diffusion-based models, step 1 has more complexity .We should note that, Several methods have recently emerged, to speed up the sampling in diffusion models[1], future extensions of our pipeline may incorporate these methods as fake generators.
>
> References:
> [1]:Zhang et al. "Fast Sampling of Diffusion Models with Exponential Integrator.
>
> 3)In principle, any unconditional diffusion model  should work in our method. Nevertheless, some proposed works (e.g. DALL.E2) do not fit our problem setting, where only images are available as the input.
>
> 4)We appreciate that there might be some implicit assumptions made in the model, but such assumptions are so general that we were able to improve upon SOTA in a wide variety of datasets including cases with medical images, natural images, and industrial images.
>
> 5)Previous methods' performance that are mentioned in our tables is largely based on what they have originally reported, and we cannot  afford to run them all to get the standard deviations. Therefore, we chose to not report the standard deviations for our method as well.
>
> 6)We have fixed the specified error and we are in the process of fixing other typographic and syntactical errors.

---

> ### Author Response · Authors · 2022-11-28
> **A friendly reminder of the rebuttal conclusion**
>
> We appreciate the valuable comments you provided. We have responded to your question and provided additional experiment results, and hope it could help address your concerns. In addition, we are more than happy to discuss and address any further questions.

---

### Author Response · Authors · 2022-11-24
**A friendly reminder of the rebuttal conclusion**

Thank you again for your thoughtful comments and constructive suggestions. We have provided thorough responses and additional experimental results. We would appreciate it if you could read our responses and update the scores if your concerns have been addressed. We are glad to further discuss any concerns that you find not fully addressed. Thank you.

Best regards,
Authors

---

### Decision · Program_Chairs · 2023-01-20

**Decision:**

Accept: poster

**Justification For Why Not Higher Score:**

All scores are weak acceptance.

**Justification For Why Not Lower Score:**

All reviewers and AC agree on the decision.

**Metareview: Summary, Strengths And Weaknesses:**

This paper studies image-based novelty detection (ND) and proposes a challenging setting, where outliers are semantically close to the samples of the normal distribution (near-ND). The authors propose to synthesise near-distribution anomalies with diffusion-based methods and train a classifier to distinguish them from the normal samples. Extensive empirical evaluations show superiority of the proposed approach in both, ND and near-ND, settings. During rebuttal the approach has been extended to even more challenging unlabeled multi-class setting, hence departing from the trend of a single-class setting and winning over one of the reviewers. AC agrees with the reviewers. AC recommends acceptance of the paper, and urges the authors to polish their final version by addressing any remaining questions, and reporting multi-class near-ND. Congratulations to the authors!

**Note From Pc:**

if the above contains the word "oral" or "spotlight" please see: "oral" presentation means -> notable-top-5% and "spotlight" means -> notable-top-25%. As stated in our emails, we are disassociating presentation type from AC recommendations